

# Microphysics of liquid water in sub-10 nm ultrafine aerosol particles

Xiaohan Li[a] and Ian C. Bourg[a,b]

[a]Department of Civil and Environmental Engineering, Princeton University, Princeton, NJ, USA
[b]High Meadows Environmental Institute, Princeton University, Princeton, NJ, USA

**Correspondence:** Ian C. Bourg (bourg@princeton.edu)

**Abstract.** Ultrafine aerosol particles with sizes smaller than 50 nm have been shown in recent studies to serve as a large source of cloud condensation nuclei (CCN) that can promote additional cloud droplet formation under supersaturation conditions. Knowledge of the microphysics of liquid water in these droplets remains limited, particularly in the sub-10 nm particle size range, due to experimental and theoretical challenges associated with the complexity of aerosol components and the small length scales of interest (e.g., difficulty of precisely sampling the liquid-air interface, questionable validity of mean-field theoretical representations). Here, we carried out molecular dynamics (MD) simulations of aerosol particles with diameters between 1 and 10 nm and characterized atomistic-level structure and water dynamics in well-mixed and phase-separated system with different particle sizes, NaCl salinities, and organic surface loadings as a function of distance from the time-averaged Gibbs dividing interface or instantaneous water-air interface. We define a sphericity factor ($\phi$) that can shed light on the phase-mixing state of nanodroplets, and we reveal an unexpected dependence of mixing state on droplet size. Our results also evidence an ion concentration enhancement in ultrafine aerosols, which should modulate salt nucleation kinetics in ultrafine sea salt aerosols, and provide detailed characterization of the influence of droplet size on surface tension and on water self-diffusivity near the interface. Analysis of water evaporation free energy and water activity demonstrates the validity of the Kelvin equation and Köhler theory at droplet sizes larger than 4 nm under moderate salinities and organic loadings and the need for further extension to account for ion concentration enhancement in sub-10 nm aerosols, droplet-size-dependent phase separation effects, and a sharp decrease in the cohesiveness of liquid water in sub-4 nm droplets. Finally, we show that an idealized fractional surface coating factor ($f_s$) can be used to categorize and reconcile water accommodation coefficients ($\alpha^*$) observed in MD simulations and experimental results in the presence of organic coatings, and we resolve the droplet-size dependence of $\alpha^*$.

## 1 Introduction

Water is a predominant component of atmospheric aerosol, with mass fraction typically larger than 70% at moderate to high relative humidities (RH > 40%), and plays crucial roles in the formation and growth of aerosol particles (Pierce et al., 2012; Karlsson et al., 2022). For example, aerosol liquid water facilitates the partitioning of gas-phase



water-soluble organic matter into the particle phase and provides a medium for aqueous reactions to form secondary aerosol particles (Faust et al., 2017; Sareen et al., 2017; Frank et al., 2020). In addition, water vapor condensation on nanoparticles can alter the phase state of aerosols and drive aerosol growth to climatically active sizes of about 80 nm and higher (Wu et al., 2018).

Because of water's abundance in atmospheric aerosols, the microphysics of aerosol water are essential to resolving
key unknowns in aerosol growth and evolution that underlie uncertainties in aerosol-cloud interactions in current weather and climate forecasts (Sato and Suzuki, 2019; Benjamin et al., 2019; Kreidenweis et al., 2019; Lawler et al., 2021). In particular, recent studies have shown that ultrafine aerosol particles with sizes smaller than 50 nm, which are abundant in the troposphere but are conventionally considered too small to affect cloud formation, can serve as an important source of cloud condensation nuclei (CCN) and promote additional cloud droplets under supersaturation
conditions (Fan et al., 2018; Williamson et al., 2019). A key challenge in understanding this phenomenon is that properties of liquid water in such ultrafine aerosol particles remain incompletely understood due to the difficulty of experimentally characterizing bulk and interfacial water in suspended nanodroplets (Ault and Axson, 2017; Bzdek and Reid, 2017; Ohno et al., 2021).

In the absence of detailed observations of aqueous chemistry in nanodroplets (Bzdek et al., 2020a), extant models
of nano-aerosol droplets rely on a simplified conceptual representation of these particles as a bulk aqueous phase, eventually with an accumulation or depletion of solutes in a negligibly-thin interfacial region. In short, water in the core of aqueous aerosol particles is treated as bulk water, and the air-particle interface is represented as the surface of bulk water, even in particles that contain only a few hundred water molecules. This simplification underlies widely used models of aerosol growth thermodynamics and kinetics, organic chemistry, water freezing, salt efflorescence,
and organic phase separation (McDonald, 1953; Laaksonen and Malila, 2016; Kulmala et al., 1997; Petters and Petters, 2016; Petters and Kreidenweis, 2007, 2008, 2013; Shiraiwa and Pöschl, 2021; Shiraiwa et al., 2012; Liu et al., 2019; Kostenidou et al., 2018; Semeniuk and Dastoor, 2020). For example, the hygroscopic growth of nano-aerosol particles is generally described within the framework of Köhler theory (McDonald, 1953; Laaksonen and Malila, 2016; Kulmala et al., 1997; Petters and Petters, 2016; Petters and Kreidenweis, 2007, 2008, 2013; Estillore
et al., 2017), which accounts for osmotic and capillary pressure effects based only on the average droplet salinity and surface curvature, neglecting any potential impacts of droplet size on water-water and water-solute molecular level interactions. Similarly, descriptions of the mass transfer kinetics of water at the particle-gas interface largely rely on the Fuchs-Sutugin approximation with a fixed empirical accommodation coefficient (Shiraiwa and Pöschl, 2021; Shiraiwa et al., 2012), which is assumed invariant with droplet phase state, droplet size, and the diffusivity
of the investigated compounds despite evidence that its value can vary by several orders of magnitude depending on experimental conditions (Liu et al., 2019; Kostenidou et al., 2018). Despite its widespread application, the conceptual simplification outlined above has not been rigorously evaluated in the case of aerosol particles. Analogous simplifications are known to be invalid in small pores and in reverse micelles, where liquid water adopts distinct properties when confined to spaces narrower than 10 nm (Bocquet and Charlaix, 2010; Bourg and Steefel, 2012).





These results suggest that water in sub-10 nm ultrafine aerosol particles may have properties distinct from those of bulk liquid water.

Molecular dynamics (MD) simulations have been extensively used to gain insight into the properties of liquid water near interfaces at the atomistic level. In the case of liquid water-air interfaces, most of these studies have focused on flat interfaces relevant to large aerosol particles (Brown et al., 2005; Thomas et al., 2007; D'Auria and

Tobias, 2009; Tang et al., 2020; Luo et al., 2020). Comparatively few have examined the properties of liquid water in sub-10 nm aerosol particles (Chowdhary and Ladanyi, 2009; Li et al., 2011; Ma et al., 2011; Lovrić et al., 2016; Sun et al., 2013; Vardanega and Picaud, 2014; Radola et al., 2017; Karadima et al., 2019; Li et al., 2013; Radola et al., 2015, 2021). These simulations have shown that interfacial water adopts distinct properties from those of bulk liquid water and that the surface tension of water droplets decreases with increasing interfacial curvature at the

nanometer scale (Bourg and Steefel, 2012; Li et al., 2010). In addition, interfacial water molecules have been shown to have highly distinct dynamics compared to bulk liquid water, including slower hydrogen bond dynamics and faster diffusion dynamics (Ni et al., 2013; Liu et al., 2004; Wick and Dang, 2005), but the apparent discrepancy between these observations and their potential implications have not been resolved. Finally, a complicating factor revealed by these simulations and by experimental observations is that the shape of nanodroplets often exhibits significant

fluctuations and is rarely a perfect sphere, which complicates the determination of interface thickness and water uptake properties (Li et al., 2016; Garrett et al., 2006).

Here, we use MD simulations to characterize nano-droplet structure, water diffusion, and water evaporation energetics in well-mixed and phase separated system with different sizes, NaCl salinities, and organic surface loadings. Our approach builds on previous MD simulation studies of nano-aerosol droplets but includes two distinct features

that have the potential to help resolve the challenges identified in previous studies. First, we use two methods of interface characterization – the time-averaged Gibbs dividing interface (Lau et al., 2015) and the instantaneous interface (Willard and Chandler, 2010) – to compare different perspectives on the properties of the interface. Second, we characterize a broad range of structural, dynamic, and energetic properties to help reveal links between these properties. We re-examine the validity of the Kelvin equation and Köhler theory and, more broadly, of treating

nano-droplet water as a bulk aqueous phase. We find that the instantaneous interface characterization enables a novel characterization of the droplet phase state and helps resolve apparent contradictions between experimental observation and MD simulation results on interfacial water dynamics. We also find that the Kelvin equation and Köhler theory accurately describe water microphysics under moderate salinity and organic loadings in nanometer-scale droplets. However, these theories underestimate water activity at droplet diameters below 4 nm and cannot

account for the droplet size effect on ion concentration enhancement in NaCl-bearing droplets and on phase separation in droplets bearing surface-active organic compounds. Overall, our results indicate that the properties of nano-aerosol water deviate strongly from those of bulk water in particles smaller than 4 nm. One implication of this finding is that molecular-scale insight is required to accurately represent the initial stages of droplet nucleation and growth.



## 2  Methods

### 2.1  Simulated systems

Molecular dynamics simulations of droplets containing 100, 500, 1000, 3000, or 5000 water molecules ($N_w$) with NaCl molality of 0, 1, 2, or 4 m were used to examine the effect of droplet size and salinity on water microphysics in nano-aerosol particles. In addition, droplets with $N_w$ = 100, 500, 1000, 3000, or 5000 with low or high organic surface loadings (0.7 or 3.3 pimelic acid molecules nm$^{-2}$, corresponding to surface loadings of 0.3 or 1.5 monolayers) were used to examine the impact of organic compounds. Sodium chloride is used as a representative inorganic salt in aerosol particles for two reasons: first, it accounts for a significant fraction of sea-spray and mineral dust aerosols, which constitute the largest mass of particulate matter emitted to the atmosphere (Lewis et al., 2004; Wise et al., 2007) and, second, relative to other inorganic ions present in aerosol particles, such as $NH_4^+$ and $SO_4^{2-}$, alkali halide salts such as NaCl have comparatively well-tested and accurate interatomic interaction potential models, the main input controlling the accuracy of MD simulation predictions (Smith and Dang, 1994; Joung and Cheatham III, 2008, 2009). Pimelic acid (PML, $C_7H_{12}O_4$) was used as a representative organic component, because dicarboxylic acids contribute a large fraction of the total identifiable resolved organic mass in fine aerosols and have been identified and quantified in both rural and urban air (Hyder et al., 2012; Kawamura and Yasui, 2005; Tran et al., 2000). Furthermore, the O/C ratio of PML (0.57) lies near the midpoint of the range commonly observed for aerosol organic materials (0.2 to 1.0) (Zhang et al., 2007; Hallquist et al., 2009; Song et al., 2018) and near the value below which liquid-liquid phase separation is commonly observed in aerosol particles ($\sim$ 0.7 to 0.8 for organic-salt-water aerosols, $\sim$ 0.6 for organic-water aerosols) (Ciobanu et al., 2009; Song et al., 2012; You et al., 2014; Renbaum-Wolff et al., 2016; Rastak et al., 2017; Song et al., 2018). These observations suggest that PML should have the potential to mimic key properties of organic substances in nano-aerosol droplets. The sizes of droplets studied in this work range from 2 to 8 nm (Appendix Table A1).

### 2.2  System preparation and MD simulations

Molecular dynamics simulations were performed with the Gromacs program (Abraham et al., 2015) using cubic simulation cells subject to periodic boundary conditions in all three directions with an edge length of at least four times the droplet diameter. Interatomic interactions were represented using the SPC/E model for water molecules (Berendsen et al., 1987), the Joung-Cheatham model for Na$^+$ and Cl$^-$ ions (Joung and Cheatham III, 2008), and the OPLS-AA model for pimelic acid molecules (Jorgensen et al., 1996). In all simulations, bonds involving H atoms were restrained using the LINCS algorithm (Hess, 2008). Simulations were carried out in the NVT ensemble with temperature maintained at 298.15 K using the Nosé–Hoover thermostat (Evans and Holian, 1985) and with the overall translational and rotational momentum of the simulated system set to zero. Coulomb and van der Waals interactions were resolved using a real space cutoff of 1.2 nm with a particle mesh Ewald sum treatment of long-range





Coulomb interactions. Each system was simulated with a time step of 1 fs for a total simulation time of 105 ns, comprised of 5 ns for equilibration and 100 ns for data analysis.

## 2.3  Droplet structure

To calculate the radius of the water droplets in most systems (except at high organic loading) the time-averaged water number density profile $\rho(r)$ was obtained for each droplet as a function of distance from the droplet center of mass. Results were fitted to the relation (Thompson et al., 1984),

$$\rho(r) = \frac{1}{2}(\rho_l + \rho_v) - \frac{1}{2}(\rho_l - \rho_v)\tanh[\frac{2}{d_i}(r - r_0)] \tag{1}$$

where $\rho_l$ is the liquid number density, $\rho_v$ is the vapor number density, and $d_i$ is the interfacial width. The density profile was used to locate the equimolar (Gibbs) dividing surface, which corresponds to a vanishing adsorption in

a one-component system, ensuring that the surface free energy per unit area so defined corresponds to the surface tension. The radius $R_e$ of the equimolar dividing surface was calculated using numerical quadrature,

$$R_e^{\ 3} = \frac{1}{\rho_v - \rho_l}\int\limits_0^\infty r^3\frac{d\rho(r)}{dr}dr \tag{2}$$

For droplets with a high organic loading, the radius was calculated instead based on the molecular volumes of water and pimelic acid:

$$R_e^{\ *} = [\frac{3}{4\pi}(\frac{N_w}{\rho_w} + \frac{N_{org}}{\rho_{org}})]^{1/3} \tag{3}$$

where $\rho_w$ is the bulk water density ($34.02 \pm 0.19$ molecules nm$^{-3}$) and $\rho_{org}$ is the bulk organic density ($4.373 \pm 0.003$ molecules nm$^{-3}$) calculated in the conditions of our simulations.

## 2.4  Diffusion coefficient

The self-diffusion coefficient of water molecules was calculated by analyzing water trajectories at 1 ps interval during

25 ns simulations for all droplets. To analyze the dependence of diffusion on position relative to the interface, we binned water molecules into different 3 Å thick layers based on their distance relative to the interface. Any molecule initially present in a specified layer that passed through the layer boundaries no longer contributed to the calculation. The self-diffusion of water molecules was investigated in directions parallel and normal to the interface.

The parallel diffusion coefficient was calculated using the Einstein relation (Liu et al., 2004)

$$D_\parallel = \frac{1}{4}\lim_{\tau \to \infty}\frac{d\langle \mathbf{l}_{\parallel,i}^2\rangle}{d\tau} \tag{4}$$

where $\mathbf{l}_{\parallel,i}^2$ is the mean-square displacement (MSD) of water molecules in directions parallel to the interface, averaged over all molecules of interest remaining in $i^{th}$ layer during the entire time interval $\tau$. The infinite-time limit in Eq. 8





was approximated by evaluating the slope of MSD vs. $\tau$ for $\tau = 2$ to 6 ps based on the linear relationship between MSD and $\tau$ on this time scale; use of longer time-scales yielded less precise results as fewer molecules remained consistently located in the i$^{th}$ layer (Figure A4).

The diffusion coefficient normal to the interface was evaluated using the anisotropic Smoluchowski equation in the r direction:

$$\frac{\partial p}{\partial t} = \frac{1}{r^2}\frac{\partial}{\partial r}[r^2 D_\perp (\frac{\partial p}{\partial r} + \beta F_r p)] \tag{5}$$

where $p = p(r, t|r_0, t_0)$ is the conditional probability distribution function, $D_\perp = D_{rr}$ is the diffusion tensor perpendicular to the interface, $F_r = -\frac{\partial W(r)}{\partial r}$ is the mean force acting on the molecule, and $W(r) = -k_B T \ln \frac{\rho(r)}{\rho_0}$ is the potential of mean force (PMF), calculated from the water density profile. $D_\perp$ values in different water layers were determined by matching the water survival probabilities calculated from MD simulation and from the numerical solution to the Smoluchowski equation as described by Wick and Dang (2005).

## 2.5 Surface tension

Surface tension was calculated by analyzing the normal component of the Irving–Kirkwood pressure tensor (Irving and Kirkwood, 1950) at 1 ps intervals during 25 ns simulations for all droplets. According to the formula given by Thompson et al. (1984), the normal pressure tensor can be written as

$$P_N(r) = P_K(r) + P_U(r) = k_B T \rho(r) + S^{-1}\sum_k f_k \tag{6}$$

where $k_B$ is the Boltzmann constant, $S$ is the surface area of the spherical surface of radius $r$, and $f_k$ is the normal component of the force between a pair of molecules acting across the surface $S$. The surface tension is evaluated as

$$\gamma \approx \frac{3W}{4\pi R_e^2} \tag{7}$$

where $W$ is the work of formation as described by Zakharov et al. (1997):

$$W = 2\pi \int\limits_0^{R_\beta} P_N(r) r^2 dr - 2\pi P^\beta R_\beta^3 / 3 \tag{8}$$

In Eq. (8), $P^\beta$ the vapor pressure and $R^\beta$ the radius of a sphere in the vapor region, which can be approximated as the equimolar radius $R_e$.

## 2.6 Free energy of evaporation

The potential of mean force (PMF) associated with the evaporation of water molecules was calculated using two methods. The first method consisted in calculating the PMF based on the water density profile during 100 ns equilibrium simulations by

$$F(r) = -RT \ln \frac{\rho(r)}{\rho_w} \tag{9}$$



The second method consisted in applying the well-known umbrella sampling technique (Torrie and Valleau, 1977), whereby a single water molecule was successively tethered at different distances from the droplet center of mass using a harmonic potential with a force constant of 1000 kJ mol$^{-1}$ nm$^{-1}$ and 100 ns of simulation time at each tethering distance. The PMF was expressed as

$$F(r) = -kT \ln P(r) + 2kT \ln r + C \tag{10}$$

where $C$ is a constant value, and $P(r)$ is defined by

$$P(r) = \int dr^N \left( \frac{e^{-\beta \Phi(r^N)}}{Z\pi(r)} \right) \pi(r) \delta(r - r(r^N)) \tag{11}$$

where $\pi(r)$ is the sampling weight and $\frac{e^{-\beta \Phi(r^N)}}{Z\pi(r)}$ is a re-weighting factor that restores the correct Boltzmann weight. The variable $\pi$ is associated with the bias potential $v(r)$ through

$$\pi(r) = e^{-\beta v(r)} \tag{12}$$

The constant $Z$ is unknown, but can be determined by

$$\int dr P(r) = 1 \tag{13}$$

The evaporation free energy of water molecules was calculated as the free energy difference of water molecules in the vapor phase and the liquid phase as

$$\Delta F = F_v - F_w \tag{14}$$

### 2.7 Accommodation coefficient

To calculate the accommodation coefficient for a single droplet, the phase-state of every water molecule in the system was recorded during a 100 ns trajectory with a timestep of 1 fs. The phase state of individual water molecules was divided into three categories (Varilly and Chandler, 2013): (1) liquid phase (state=-1): any water molecule that is
hydrogen-bonded to at least one water molecule and is at most 4 Å away from the nearest water molecules (where the "position" of a water molecule refers to the position of its center of mass unless otherwise stated); (2) vapor phase (state=1): any water molecule that has no hydrogen-bonded water neighbors and is more than 8 Å away from its nearest water neighbor; (3) intermediate phase (state=0): any water molecule that is neither in liquid nor vapor phase. An accommodation event is identified if the state of a water molecule changes from 1 at timestep $t_1$ to -1 at
timestep $t_2$, without adopting a non-zero state value during the time interval $(t_1, t_2)$. A reflection event is identified if the state of a single water molecule changes from 1 at $t_1$ to 1 at $t_2$, and stays in state 0 during the time interval $(t_1, t_2)$. The accommodation coefficient $\alpha$ was calculated as

$$\alpha = \frac{N_{acc}}{N_{acc} + N_{ref}} \tag{15}$$





where $N_{acc}$ is the number of accommodation events and $N_{ref}$ is the number of reflection events. The reported
normalized accommodation coefficient $\alpha^*$ was calculated as $\alpha^* = \alpha/\alpha_{0,w}$, where $\alpha_{0,w}$ (the value for the flat surface
of pure water) was approximated as the $\alpha$ value of the largest simulated pure water droplet with $N_w = 5000$.

## 3 Results and discussions

### 3.1 Morphology and phase state characterization

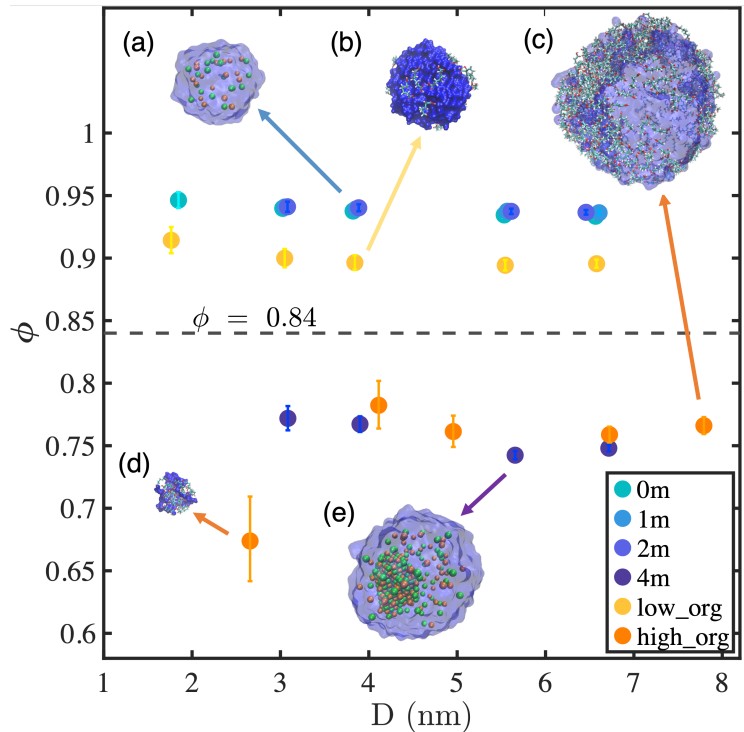

**Figure 1.** Sphericity factor ($\phi$) of nanodroplets with different compositions as a function of droplet diameter. Simulation
snapshots are shown for the droplets with water (blue color), Na$^+$ (light orange spheres), Cl$^-$ (green spheres), and pimelic
acid (colored chains): (a) 1 m NaCl with $N_w = 1000$; (b) low organic surface loading with $N_w = 1000$; (c) high organic surface
loading with $N_w = 5000$; (d) high organic surface loading with $N_w = 100$; and (e) 4 m NaCl with $N_w = 3000$.

Insets (a)-(e) in Figure 1 show representative snapshots of simulated droplets that illustrate the variety of predicted
morphologies and phase mixing states. Additional droplet snapshots are presented in Appendix Figure A1. For NaCl
droplets, Na$^+$ and Cl$^-$ ions are repelled from the water-air interface and distributed inside the droplets in all
simulations. These ions are crystallized in the center of the droplet at 4 m concentration, forming a phase state
illustrated in Figure 1(e), whereas they are well-mixed with water molecules at 1 m and 2 m concentrations with



phase state illustrated in Figure 1(a). The NaCl solubility implied by our results (between 2 and 4 m) is consistent
with that observed in previous MD studies that examined bulk NaCl solutions using the same interatomic potential
(Mester and Panagiotopoulos, 2015a, b). For droplets with PML, the arrangement of organic molecules varies with
droplet size and organic surface loading. At low organic loading, PML molecules are predominantly adsorbed at the
water-air interface with minimal clustering, as illustrated in Figure 1(b). At high organic loading, when droplet size is
sufficiently large (diameter D ≥ 4.1 nm), PML molecules retain their affinity for the water-air interface and cluster at
the droplet surface, forming the 'engulfed' morphology shown in Figure 1(c). In smaller droplets with a high organic
loading, however, the organic molecules cluster in the center of the droplet and displace water molecules towards
the surface as shown in Figure 1(d). This transition suggests the existence of a threshold, at a droplet diameter of
3 to 4 nm, in the competition between water hydrogen bonding, which promotes water clustering, and hydrophobic
attraction between PML molecules, which promotes organic clustering. Our results are consistent with previous
findings that the hydrogen bond network formed by bulk liquid water involves cooperativity over length scales of
1 to 2 nm and becomes significantly weaker upon attenuation of long-range water-water orientational correlations
(Alper and Levy, 1989; Ohmine and Tanaka, 1993).

To characterize the various phase states identified above, we propose a quantitative variable, referred to hereafter
as the sphericity factor ($\phi$), defined as the ratio of the surface area of a perfect spherical water droplet, $S_{\text{w,sp}}$, to the
instantaneous interface surface area formed by water molecules with the same amount of water, $S_{\text{w,inst}}$:

$$\phi = \frac{S_{\text{w,sp}}}{S_{\text{w,inst}}} \tag{16}$$

The $\phi$-value is in the range of (0, 1] and reflects the tendency towards water clustering. In particular, $\phi = 1$ indicates
that water forms a perfect spherical droplet with no capillary waves or disruption due to phase separation.

Figure 1 presents the $\phi$-values of all simulated systems as a function of droplet diameter $D$. By visual inspection
of our simulation trajectories, we identify droplets as being in a well-mixed state if no single phase is formed except
for the water cluster and in a phase separated state if an additional phase exits. For the conditions examined in
this study, a threshold at $\phi \approx 0.84$ (illustrated by the horizontal gray dashed line in Figure 1) distinguishes between
well-mixed (above the threshold) and phase separated droplets (below the threshold). For well-mixed systems, at
fixed ion concentration or organic surface excess, $\phi$ decreases slightly with increasing droplet size, indicating that
larger droplets have a more fluctuating water-air interface. This observation is consistent with our results showing
that the water-air interface width observed in time-averaged density profiles increases with droplet size (Appendix
Table A1) and agrees with the expectation that the capillary wave contribution to interfacial fluctuation increases
with interfacial area (Ismail et al., 2006; Lau et al., 2015). We also find that $\phi$ increases with ion concentration
but decreases with organics at the interface, which indicates that NaCl reduces, but PML enhances, interfacial
fluctuations. This observation is consistent with the variations in interfacial width with salinity and organic surface
loading (Appendix Table A1) and with the expectation that NaCl and PML should respectively increase or decrease





surface tension because of their negative (NaCl) or positive (organic) adsorption at the water-air interface, as confirmed below.

Phase-separated systems (i.e., droplets with either 4 m NaCl or a high organic loading) also show a small decrease in $\phi$ with increasing droplet size in most cases. A significant exception is presented by the smallest droplet with a high organic loading, which shows a sharp decrease in $\phi$ with *decreasing* droplet size between D = 4.1 nm and 2.7 nm. This decrease in $\phi$ coincides with a change in droplet phase-mixing state from a water-core structure at $D \geq 4.1$ nm to an organic-core structure at $D = 2.7$ nm and shows that the phase-mixing state of nanodroplets can abruptly vary with particle size. Organic particles with sizes below 5 nm and different organic-water ratios, which lie in the above mentioned phase-mixing transition region and may shed light on the early stages of hygroscopic growth of secondary organic aerosols (Qin et al., 2021), will be examined in more detail in an upcoming study.

## 3.2 Interfacial structure of nanodroplets

To gain additional insight into the phase state behaviors outlined above, the radial density profiles of water, ions, and PML molecules in our nano-droplets were calculated relative to either the time-averaged interface (Gibbs dividing surface) (Lau et al., 2015) or the instantaneous interface (Willard and Chandler, 2010). As shown in previous studies of flat water-air interfaces (Willard and Chandler, 2010) and in Appendix Figure A2, the atomic density profile obtained under the instantaneous interface scheme provides more detailed information about molecular structure at the interface (e.g., water and solute density layering). Figure 2(a) shows ion density relative to the instantaneous interface in 1 m NaCl droplets with different water molecules. Our results show that both $Na^+$ and $Cl^-$ ions are negatively adsorbed at the water-air interface, but $Na^+$ is more strongly excluded from the interface than $Cl^-$, such that the interface is negatively charged, in agreement with previous studies of flat water-air interfaces (Jungwirth and Tobias, 2000; Underwood and Greenwell, 2018) (see Appendix Figure A2 for more density profiles). Although all droplets highlighted in Figure 2(a) were initialized as 1 m uniformly mixed NaCl solutions, ion concentrations are strongly enhanced (up to >2 times the initial concentration) in the core of the droplets at equilibrium due to the negative adsorption of ions at the water-air interface. This ion concentration enhancement increases with decreasing droplet size, particularly in droplets smaller than 4 nm ($N_w < 1000$). We note that the interface propensity of halide anions varies with anion size and polarizability and that the simple interatomic potential models used here may underestimate the affinity of halide anions for the interface (Levin et al., 2009; Jungwirth and Tobias, 2006; Caleman et al., 2011). However, the negative adsorption of alkali cations predicted here is well-established (Underwood and Greenwell, 2018; Levin et al., 2009; Jungwirth and Tobias, 2006; Caleman et al., 2011), and the overall negative adsorption of NaCl salt at the water-air interface is correctly predicted as shown by previous MD simulation results on the salinity-dependence of water-air interfacial tension (Jungwirth and Tobias, 2001, 2006; D'Auria and Tobias, 2009; Horinek et al., 2009; Sun et al., 2012).

To illustrate the expected effect of droplet size on ion concentration, we define a salt concentration enhancement factor ($\epsilon$) as the ratio between salt concentration in the core of the droplet and average salt concentration in the



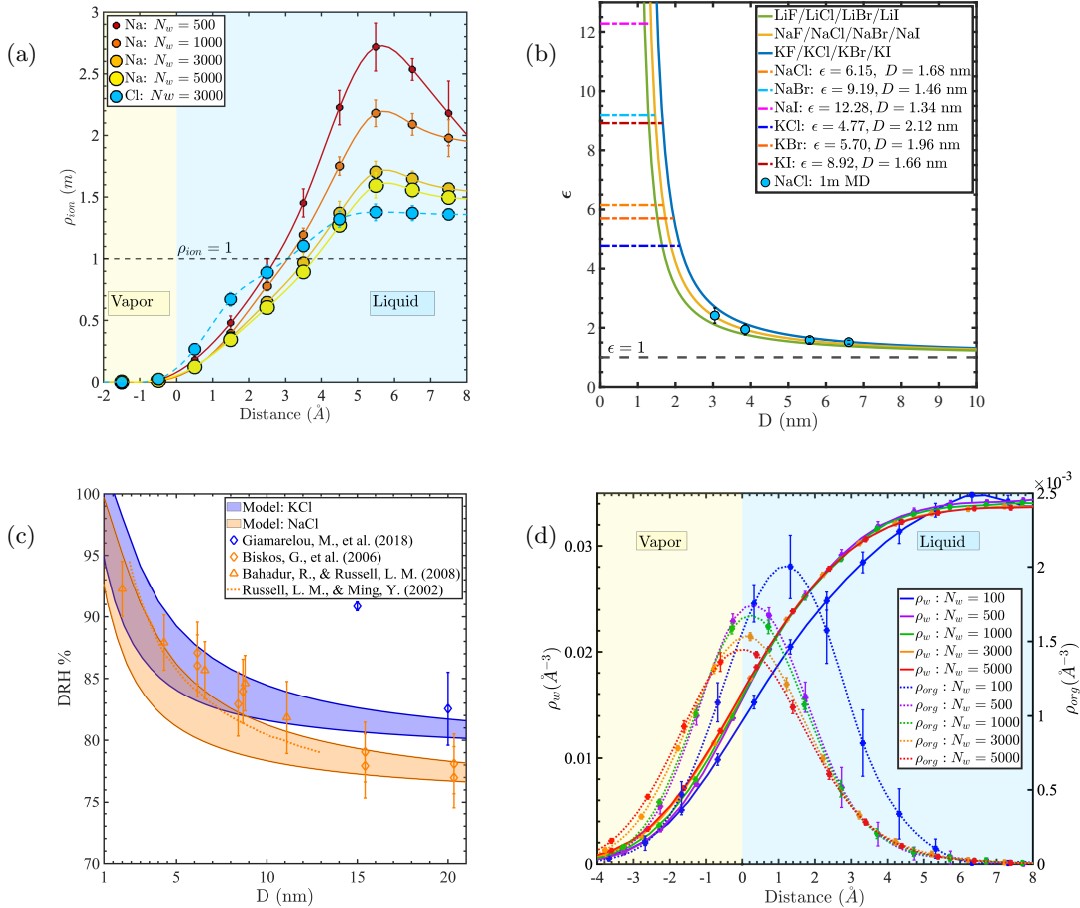

**Figure 2.** (a) Density profiles of $Na^+$ and $Cl^-$ ions ($\rho_{ion}$) relative to the instantaneous interface in 1m NaCl droplets with different number of water molecules. (b) Salt concentration enhancement factor $\epsilon$ as a function of droplet diameter $D$ for droplets with an average salinity of 1 m (blue circles: MD simulation results; lines: predictions for different salts based on a simplified model whereby the salt exclusion distance at the water-air interface equals 1.2 times the first hydration radius of the cation; horizontal dashed lines indicate the $\epsilon$ values where different salts would be expected to precipitate, based on their solubility in bulk liquid water, in a droplet with average salinity of 1 m. (c) Deliquescence relative humidity (DRH) of KCl and NaCl nanoparticles as a function of diameter (D): shaded area are model predictions obtained in this study; blue and orange diamonds are previous experimental results for KCl and NaCl (Biskos et al., 2006a; Giamarelou et al., 2018); orange triangles are previous MD simulation results for NaCl (Bahadur and Russell, 2008); and the orange dashed line is the NaCl DRH prediction based on a thermodynamic model (Russell and Ming, 2002). (d) Atomic density profiles of water molecules ($\rho_w$) and PML molecules ($\rho_{org}$) relative to the Gibbs dividing surface under the time-averaged interface scheme at low organic loading.





entire droplet. Predicted $\epsilon$-values for different alkali-halide salts are shown in Figure 2(b) based on the simplifying assumption that the salt is excluded from the vicinity of the water surface, in a region with a thickness equal to 1.2 times the first hydration radius of alkali cations, quantified as the first minimum of the ion-water radial distribution function predicted using the SPC/E water model and compatible ions models (Joung and Cheatham III, 2009).

Our simplistic model, shown as solid curves in Figure 2(b) for different salts, is consistent with our MD simulation predictions (blue circles, shown for our systems with 1 m NaCl), indicating that the length scale associated with salt exclusion at the water-air interface is invariant with surface curvature. Horizontal lines in Figure 2(b) show the $\epsilon$-values necessary to achieve crystallization of different salts based on their experimental solubility (Joung and Cheatham III, 2009) in droplets where the average concentration is 1 m. Overall, both our simplistic model and our

detailed atomistic simulation predictions indicate that the concentration enhancement of simple alkali-halide salts is well above 1.25 for sub-10 nm droplets, and may be more than 10 for droplets with a diameter below 1.5 nm. In other words, in droplets with common sea-salt ionic species, crystallization should occur more readily than expected based on the average salt concentration, particularly at small droplet sizes. Our results suggest that the tendency for salt deliquescence should be dimished and salt efflorescence should be enhanced in ultrafine nanoparticles. The

ion concentration enhancement discussed here is consistent with experimental findings that both the deliquescence and efflorescence relative humidities (DRH, ERH) of NaCl and KCl particles increase as droplet size decreases from 60 nm to less 10 nm, with a sharper decrease at smaller droplet sizes (Biskos et al., 2006a, b; Giamarelou et al., 2018; Hämeri et al., 2001). Shaded areas in Figure 2(c) show the KCl and NaCl DRH as a function of nanoparticle diameter predicted from the ion concentration enhancement model discussed above (see Appendix B for calculation

details). Our predictions are consistent with both the size dependence of NaCl DRH reported in previous studies (Biskos et al., 2006a; Bahadur and Russell, 2008; Russell and Ming, 2002) and with previous observations that KCl has a larger DRH than NaCl (Giamarelou et al., 2018). Overall, the concentration enhancement discussed above may help explain experimental observations of a size-dependence of aerosol shape, in which nano-aerosol particles generated from well-controlled sea spray chambers are more cubic at smaller sizes but more spherical at larger sizes

(Zieger et al., 2018). Our observations indicate that the $\epsilon$ factor becomes particularly significant for sub-10 nm particles and should be taken into consideration in predictions of salt microphyics in ultra-fine particles.

Figure 2(d) shows the density profiles of water and PML molecules based on the location of their center of mass under the time averaged interface scheme at low organic surface loading. The PML density profiles show a sharp peak near the water-air interface, confirming the high affinity of the organic compound for the interface. The

location of the PML density peak shifts into the liquid water region with decreasing droplet size, particularly at sizes below 3 nm ($N_w < 500$). This observation suggests that the hydrophilic character of the water phase decreases with decreasing number of water molecules in the droplet, as already noted above in the case of the smallest droplet at high organic loading. The PML density curves at high organic loading (Appendix Figure A3(d)) reveal a more complex and variable pattern of organic-water mixing phase states, in agreement with the transition between water-

and organic-core morphologies identified above.





## 3.3 Water diffusion kinetics in nanodroplets

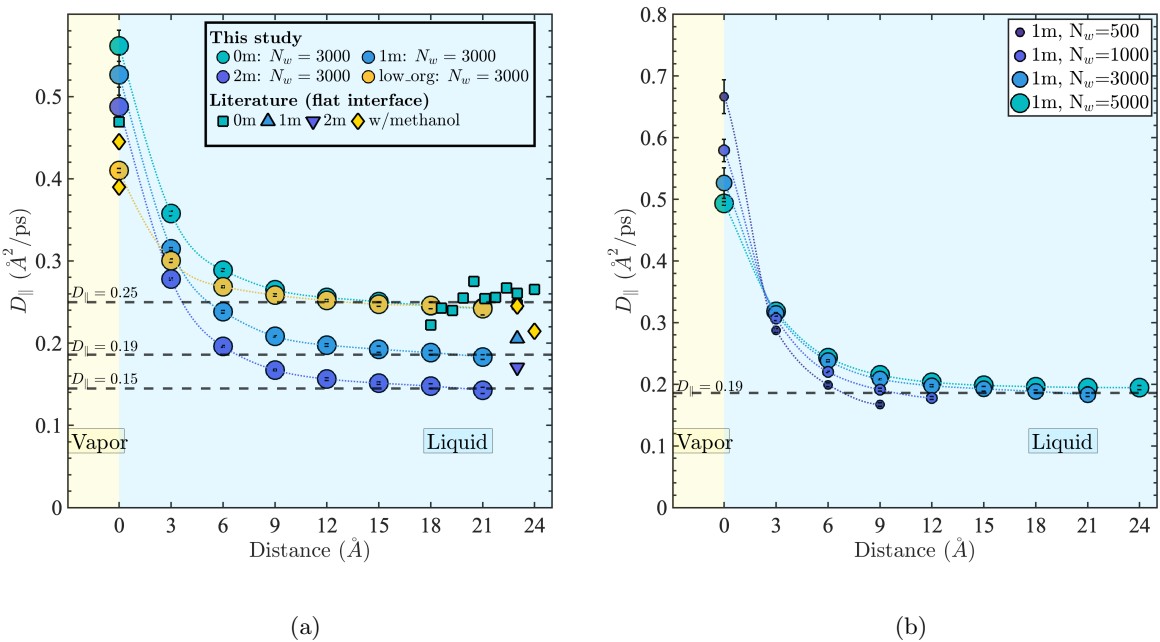

**Figure 3.** Water diffusion coefficient in the direction parallel to the interface ($D_\parallel$) as a funtion of distance from the time-averaged interface ('com' scheme) (a) in 5.5 nm droplets ($N_w = 3000$) with different aqueous chemistries and (b) in 1 m NaCl droplets with different water contents (symbol sizes are scaled with droplet diameter). Literature data shown in (a) were collected from previous MD simulation studies using the SPC/E water model: 0 m (Kim et al., 2012; Paul and Chandra, 2005; Tsimpanogiannis et al., 2019); 1 and 2 m (Kim et al., 2012); and methanol-water solution at low organic concentrations (w/methanol) (Paul and Chandra, 2005). All literature $D_\parallel$ values were obtained for systems with a flat water-air interface.

Water diffusion in nanodroplets is a fundamental phenomenon that influences interfacial mass fluxes and chemical transformation rates (Chan et al., 2014; Slade and Knopf, 2014; Julin et al., 2013; Evoy et al., 2020). Details of this phenomenon are complicated by the inherently anisotropic and non-uniform nature of water diffusion near the interface. In this study, the water self-diffusion coefficient is calculated in discrete 3 Å thick water layers in directions parallel ($D_\parallel$) or perpendicular ($D_\perp$) to the interface. As explained in the Methods section, we calculated $D_\parallel$ using the Einstein relation and $D_\perp$ using the anisotropic Smoluchowski equation (Liu et al., 2004). Diffusion coefficients were calculated relative to the center of mass of the droplet ('com' scheme) or to the instantaneous interface ('inst' scheme). As expected, both schemes yield identical results for $D_\parallel$, while for $D_\perp$ the 'com' scheme yields larger values due to the presence of capillary waves at the interface (Appendix A2). The following discussion focuses primarily on our predicted $D_\parallel$ values. Our $D_\perp$ values are somewhat less precise and are discussed in the supplementary information (Appendix A2.2).





Figure 3(a) shows $D_\parallel$ in different simulated systems relative to the distance from the Gibbs diving surface. Our results show that $D_\parallel$ values are bulk-liquid-like at distances $> 9$ Å (i.e. $> 3$ water layers) from the interface for all

well-mixed systems. The $D_\parallel$ values at the interface are 2 to 3 times larger than in the core of the droplet, with potential implications for mass transfer and reaction kinetics at the interface. In addition, $D_\parallel$ in the droplet core decreases from 0.25 Å$^2$ ps$^{-1}$ to 0.15 Å$^2$ ps$^{-1}$ as ion concentration increases from 0 m to 2 m, while $D_\parallel$ at the interface decreases with increasing organic surface loading. The $D_\parallel$ difference between interfacial layer and bulk liquid region, and the $D_\parallel$ change with salinity and organic addition observed in our results are consistent with the trend reported

in previous studies (Kim et al., 2012; Paul and Chandra, 2005; Tsimpanogiannis et al., 2019) as shown in Figure 3(a). In all well-mixed NaCl droplets (1, 2 m), we consistently observe a droplet size dependence of water diffusion near the interface as illustrated in Figure 3(b). More precisely, $D_\parallel$ at the interface decreases with droplet size, while $D_\parallel$ in the few monolayers below the interface increases with droplet size. This observation is consistent with our results showing that for droplets containing water and NaCl, smaller droplets have lower ion density at the interface

and greater ion density in the droplet core. The minor offset between our $D_\parallel$ values in the bulk-liquid-like region and those reported by Kim et al. (2012) at the same salinity (calculated in bulk liquid NaCl solutions) as shown in Figure 3(a), is also consistent with the ion density enhancement in the droplet core noted above. Finally, the faster water diffusivity at the interface observed in the case of $D_\parallel$ is not observed in the case of $D_\perp$, as discussed further in Appendix A2. In fact, results obtained under the 'inst' scheme yield $D_\perp \approx 0.15$ to $0.17$ Å$^2$ ps$^{-1}$ near the interface,

i.e., water diffusivity normal to the interface is smaller than in the droplet core, in agreement with studies showing slower reorientational dynamics of water near hydrophobic surfaces (Laage et al., 2009; Fayer et al., 2010; Fayer, 2012).

### 3.4    Surface tension of nanodroplets

Surface tension is an important factor that governs the hygroscopic growth and the activation of aerosol particles

and is incorporated in climate models through Köhler theory. However, the surface tension of nano-aerosol particles, especially at the sub-10 nm scale, is not well known and remains arduous to characterize experimentally (Noziere et al., 2014; Bzdek et al., 2016, 2020b; Lee and Tivanski, 2021). Appendix Figure A10 shows surface tensions ($\sigma$) calculated for our different droplets based on the Irving-Kirkwood pressure tensor (Irving and Kirkwood, 1950; Thompson et al., 1984). A representative example of the instantaneous normal pressure tensor is shown in Appendix

Figure A9. All calculated $\sigma$ values are also tabulated in Appendix Table A1. We note that $\sigma$ values reported in different MD simulation studies differ significantly even for pure liquid water system (by up to $\approx 20$ mN m$^{-1}$), depending on the interatomic potential models and surface tension determination methods employed (Bahadur et al., 2007; Bahadur and Russell, 2008; Li et al., 2010; Lau et al., 2015). For well-mixed NaCl droplets with 0, 1, and 2 m NaCl concentrations, our results show that surface tension increases with droplet size and ion concentration

and approaches to 55, 56, and 58 mN m$^{-1}$ respectively when $N_w \geq 3000$ (i.e., $D > 5$ nm). The predicted increase in surface tension with NaCl salinity is consistent with experimental values reported for flat water-air interfaces (1.3





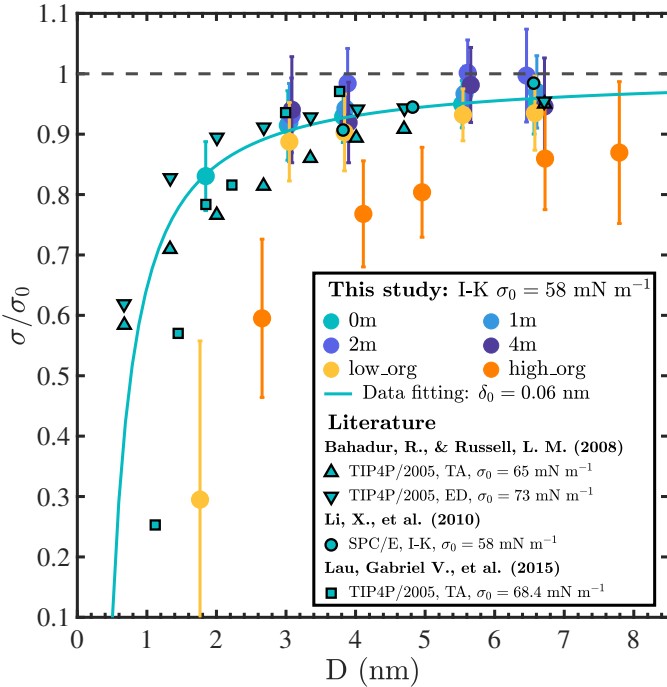

**Figure 4.** Normalized surface tension ($\sigma/\sigma_0$) of droplets as a function of droplet diameter (D). Colored circles show results obtained in this study using the SPC/E water model and the Irving-Kirkwood (I-K) pressure tensor method; symbols with black edge show previous MD simulation results obtained for pure water droplets at ambient temperature using the TIP4P/2005 or SPC/E water models and the test-area (TA), energy difference (ED) or I-K methods (Bahadur and Russell, 2008; Li et al., 2010; Lau et al., 2015); the light blue solid line shows a fit of our 0 m surface tension data to the second-order Helfrich expansion, which yields the Tolman length $\delta_0 = 0.06$ nm.

to 1.8 mN m$^{-1}$ per 1 m of salinity) (Hård and Johansson, 1977; Washburn et al., 1926; Abramzon and Gaukhberg, 1993; Tuckermann, 2007). To highlight the impacts of curvature, salinity and organic coatings on $\sigma$ and compare our results with those of other studies, we reported normalized surface tension $\sigma/\sigma_0$ as well as corresponding $\sigma_0$ values

in Figure 4, where $\sigma_0$ is the surface tension value reported for the flat pure liquid water-air interface in the same study. Our $\sigma_0$ value is 58 mN m$^{-1}$, in agreement with previous estimates of the surface tension of SPC/E water (Ismail et al., 2006). The solid light blue line in Figure 4 shows the fit of our $\sigma/\sigma_0$ results for 0 m droplets to the second-order Helfrich expansion, i.e. $\sigma/\sigma_0 = 1 - 2\delta_0/r + c/r^2$, where $\delta_0$ is the Tolman length, $r$ is the droplet radius, and $c$ is the rigidity coefficient (Wilhelmsen et al., 2015). The best fit is achieved using $\delta_0 = 0.06$ nm (95% confidence

interval [0.032, 0.094] nm) and $c = -0.025$ nm$^2$ (95% confidence interval [-0.096, 0.045] nm$^2$). The positive value of $\delta_0$ in this study, which indicates that $\sigma$ increases with droplet size, agrees with the trends observed in other studies as shown in Figure 4. In addition, although the absolute $\sigma$ values reporting in different studies differ depending on the choice of simulation methods, $\sigma/\sigma_0$ data from previous studies (Bahadur and Russell, 2008; Li et al., 2010; Lau et al.,




2015) agree well with our simulation results and collapse near the light blue fitting curve. This observation shows
that, for pure water droplets, $\sigma$ decreases sharply as droplet size decreases below 4 nm, and can even be less than
half of $\sigma_0$ at sizes below 1 nm. For droplets with an organic surface loading, surface tension shows the same trend
with particle size as for pure water droplet, but also decreases with organic surface loading for all systems where the
droplet has a water core ($N_w \geq 500$), in agreement with experiment observations that water surface tension decreases
with organic adsorption at the water-air interface (Tuckermann, 2007; Kiss et al., 2005). An exception is observed in
Figure 4 in the case of the smallest droplets with low and high organic loadings (left-most yellow and orange circles
in Figure 4), which both contain the same amount of water ($N_w = 100$), yet the droplet with high organic loading
has a higher surface tension. The surface tension of the organic-core configuration (i.e., the smallest droplet with
a high organic loading) is close to that observed for pure droplets of PML (on the order of 30 mN m$^{-1}$ at $D \approx 3$
nm), suggesting that the unexpected increase in surface tension with organic loading is associated with the observed
morphology reversal from a water core to an organic core. In short, the common assumption in aerosol microphysics
studies that surface tension invariably decreases with increasing organic loading in aqueous aerosols appears invalid,
at least in some cases, for droplets smaller than 4 nm. This finding may have important implications, in particular,
in understanding the initial nucleation and growth of secondary organic aerosols (Enghoff and Svensmark, 2008;
Kürten et al., 2018; Guo et al., 2020).

## 3.5 Validity of the Kelvin equation

The evaporation free energy $\delta F$, which is the free energy difference associated with transferring a single water
molecules from the droplet phase to the vapor phase, is an important factor that governs water evaporation kinetics
through the relation (Seinfeld and Pandis, 2008):

$$j \approx c \exp\left(-\frac{\delta F}{RT}\right) \tag{17}$$

where $j$ is the evaporation rate, $c$ is a constant factor, $R$ is the ideal gas constant, and $T$ is absolute temperature.
From classical thermodynamic Kelvin theory, the water evaporation free energy for a droplet with radius $r$, $\delta F(r)$,
relates to the value for a bulk liquid system with a flat interface and with the same aqueous chemistry, $\delta F_b$, through
the relation

$$\delta F(r) - \delta F_b = -RT \ln \frac{P_v(r)}{P_{v,b}} = -\frac{2\sigma V_m}{r} \tag{18}$$

where $P_v(r)$ is the saturated vapor pressure of the droplet, $P_{v,b}$ is the saturated vapor pressure of the bulk liquid
system with a flat interface and the same aqueous chemistry, $\sigma$ is the surface tension of the droplet, and $V_m$ is the
molar volume of the liquid. Therefore, if the Kelvin equation is valid and if solute concentration is invariant with $r$,
a plot of $\delta F$ as a function of $2\sigma/D$ (where $D = 2r$) should yield a straight line with a slope of $-2V_m$. Such a plot is
presented in Figure 5(a).



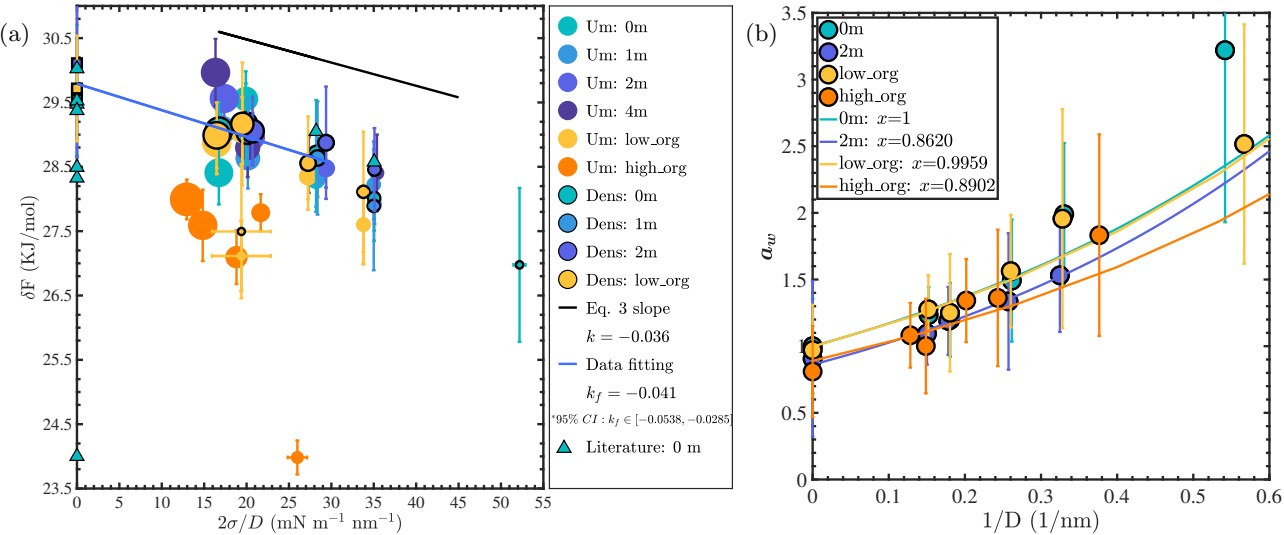

**Figure 5.** (a) Water evaporation free energy calculated from the density profile in equilibrium MD simulations ('Dens') or umbrella sampling method ('Um'): the black line show the reference slope of $\delta F$ vs $2\sigma/D$ from the classical Kelvin equation prediction, the blue line show the best linear fit obtained for well-mixed NaCl systems with sizes larger than 3.5 nm ($N_w \geq 1000$). Marker size is scaled with the droplet size. Literature data were collected from previous MD simulations using the SPC/E water model (Garrett et al., 2006; Li et al., 2018; Varilly and Chandler, 2013). (b) Water activity $a_w$ (circles) versus $1/D$ for different droplets: colored lines show the theoretical $a_w$ predictions from Köhler theory.

410    For each simulated system, we calculated $\delta F$ using two different approaches. The first method consisted in determining the free energy of water evaporation using the umbrella sampling method (Torrie and Valleau, 1977), a biased MD simulation method in which a single water molecule is progressively steered from the particle to the vapor phase. The second method consisted in determining the equilibrium water density in the liquid and vapor phases, $\rho_l$ and $\rho_v$, from which the evaporation free energy was obtained as $\delta F = -RT \ln \rho_v/\rho_l$. The second method

415    was applied only for systems where a well-defined aqueous phase (and corresponding $\rho_l$ value) could be identified. From Figure 5(a), we observe that the umbrella sampling and density profile methods yield consistent $\delta F$ values for all well-mixed systems (0 m, 1 m, 2 m and low organic loading). We note that there are limited $\delta F$ literature data even for pure water system. Even for MD simulations of the flat water-air interface carried out with the same water model as in the present study (SPC/E), $\delta F$ values reported in previous studies vary significantly, as shown

420    in Figure 3(a) (Garrett et al., 2006; Li et al., 2018; Varilly and Chandler, 2013). The comprehensive $\delta F$ dataset reported in our study, with longer simulation times, multiple analysis approaches, and a range of droplet sizes and water chemistry conditions, provide a useful guide for future examinations of the evaporation free energy of water molecules from aerosol particles.





Calculated $\delta F$ values in our study increase with droplet size (i.e., water evaporates more readily from smaller droplets) as expected from Eq. 3. Although the impact of ion concentration on evaporation energy cannot be precisely resolved, a linear relationship between $\delta F$ and $2\sigma/D$ is clearly observed in the case of well-mixed droplets, in agreement with the Kelvin equation. The theoretical slope for pure water droplets according to the Kelvin equation prediction ($k = -36 \times 10^{-6}$ m$^3$ mol$^{-1}$, calculated using the molar volume of liquid water predicted by the SPC/E water model) is indicated by the black line in Figure 5(a). This theoretical slope lies within the 95% confidence interval of the slope obtained by linear regression of the data for well-mixed NaCl droplets with sizes larger than 3.5 nm ($N_w \geq 1000$) (blue solid line in Figure 5(a) with slope $k_f = -41 \pm 13 \times 10^{-6}$ m$^3$ mol$^{-1}$). However, if the evaporation data obtained for droplets smaller than 3 nm ($N_w \leq 500$) are included in the regression, the best fitting line is deflected downward and $k$ falls outside of $k_f$'s 95% confidence interval.

For droplets with a low organic loading, the trend shown in Figure 5(a) is generally consistent with that observed for well-mixed water and NaCl droplets. However, the smallest low organic loading droplet represents a clear outlier in Figure 5(a), as also observed in Figure 4, suggesting a possible link between its low surface tension and its low $\delta F$ value. For the 4 m NaCl droplet system, the trend in $\delta F$ (dark purple circles) is similar to that observed for the well-mixed NaCl systems, suggesting that although the ions in center of the droplet form a solid, the water surface is similar to that of the well-mixed ion-water systems. In contrast, for the phase separated high organic loading system (orange circles), predicted $\delta F$ values are consistently much smaller than those of the well-mixed systems. This indicates that at high organic loading, the organic coating strongly enhances water evaporation and invalidates Eq. 3. For the smallest droplet with a high organic loading, $\delta F$ is only 24 kJ mol$^{-1}$, deviating significantly even from the values obtained for larger droplets with high organic surface loading, an observation that may be related to the reversal from water-core to organic-core morphology identified above.

To conclude, the Kelvin equation accurately predicts water evaporation energetics for salty NaCl droplets with sizes larger than 3.5 nm, even at salinities up to 4 m where salt precipitation is observed. However, it overestimates the stability of droplets smaller than 3 nm and of droplets with a high organic surface loading, in agreement with the expectation that water becomes a somewhat distinct (less hydrophilic) fluid when the collective nature of the water hydrogen bond network is disrupted.

## 3.6 Validity of Köhler theory

Water activity $a_w$, which is the ratio of the partial pressure of water in solution and the saturated water vapor pressure of bulk pure water, is an important factor that governs the hygroscopic growth and mass concentration of aerosol particles at a given relative humidity (Seinfeld and Pandis, 2008). According to Köhler theory (Seinfeld and Pandis, 2008), the water activity of an aerosol droplet can be expressed as

$$a_w = P_v(r)/P_{v,0} = x_w \exp \frac{2\sigma V_m}{RTr} \tag{19}$$





where $x_w$ is the effective mole fraction of water molecules in the solution, which accounts for the effect of solutes as represented using Raoult's law, and $P_{v,0}$ is the saturated vapor pressure of pure water system with a flat interface. To evaluate Köhler theory at the nanometer scale, we calculated $a_w$ in all simulated droplet systems and compared our results with Eq. 4 as shown in Figure 5(b). For each system, we calculated the water activity as $a_w = \rho_v/\rho_{v,0}$,

where $\rho_{v,0}$ is the density of water in the vapor phase in a system with pure water and a flat water-air interface. The theoretical prediction was determined by fitting $a_w$ values obtained for droplets with sizes larger than 3.5 nm using Eq. 4, with $\sigma$ and $V_m$ obtained from simulations of systems with bulk-liquid-like water and a flat water-air interface (with the same salinity or organic surface loading) and $x_w$ fitted to minimize the root mean square deviation from simulation results. From Figure 5(b), we see that $a_w$ increases with decreasing droplet size. For pure water

droplets, within the precision of our results, the simulation data match the theoretical prediction well for droplets larger than 3.5 nm in diameter. For droplets smaller than 3 nm, water activity is underestimated by Köhler theory, indicating that small nano-droplets are less stable than expected (i.e., they exhibit higher saturated vapor pressure than expected, as noted above). For 2 m NaCl droplets, within the precision of our results, the simulation results are consistent with Eq. 4. However, we note that under the ideal solution assumption that underlies Köhler theory,

2 m NaCl droplet should have $x_w = 0.93$, whereas our result are consistent with Eq. 4 with an effective water mole fraction of $x_w = 0.86 \pm 0.02$, which indicates that the impact of NaCl on the activity of water is roughly twice as large as predicted by Raoult's law. Since the interatomic potential models used in this study accurately represent experimental water activity in bulk liquid water up to 3 m NaCl salinity (Mester and Panagiotopoulos, 2015a), this observation likely relates to the presence of the water-air interface. The magnitude of the observed deviation is

consistent with the ion concentration enhancement effect discussed above. For droplets containing organic matter, as the mixing state of water and organic matter is often poorly defined, the water effective mole fraction is often fitted to match measured values of $a_w$ (Seinfeld and Pandis, 2008). From our fitting results, the effective $x_w$ value in our low organic loading system equals $0.996 \pm 0.006$, indicating that at low surface coverage PML has essentially no impact on water activity. In contrast, in our high organic loading systems, the effective $x_w$ value equals $0.890 \pm 0.011$, indicating

that PML can lower water activity by 10% in engulfed structure droplets. Overall, our results are consistent with Köhler theory for droplets with sizes larger than 3.5 nm. However, Köhler theory overestimates the stability of water in droplets smaller than 3 nm (a phenomenon that may be related to the lower surface tension and water cohesion in small droplets identified above) and underestimates the impact of ions on water activity (a phenomenon that may be related to the ion concentration enhancement noted above).

## 3.7 Accommodation coefficient of water and condensation kinetics

The mass accommodation coefficient of water molecules on nano-aerosol particles, defined as the probability that a water vapor molecule that impinges on the particle surface becomes incorporated in the particle, is a fundamental factor that governs the hygroscopic growth kinetics of atmospheric aerosols (Diveky et al., 2021; Seinfeld and Pandis, 2008). Its precise value in atmospheric conditions remains highly uncertain (within a range of 0.01 to 1) despite




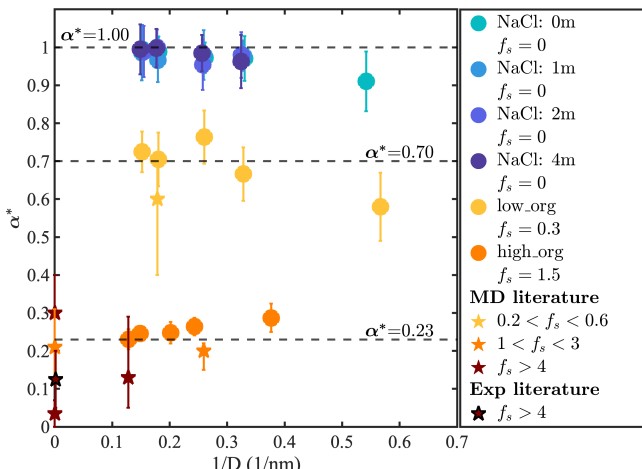

**Figure 6.** Normalized water accommodation coefficient $\alpha^*$ versus $1/D$ for different droplets. Circles show results calculated in this study. Stars show previous MD simulation (Ma et al., 2011; Chakraborty and Zachariah, 2008; Li et al., 2019; Takahama and Russell, 2011) and experimental results (Diveky et al., 2019; Duffey et al., 2013; Diveky et al., 2021) obtained at ambient temperature categorized by their $f_s$ values.

decades of study because of its complex dependence on particle composition, phase-mixing state, and surface curvature (Diveky et al., 2021; Shiraiwa and Pöschl, 2021; Barclay and Lukes, 2019; Davis, 2006; Winkler et al., 2004). In our study, the water accommodation coefficient ($\alpha$) was calculated by tracking the trajectory and phase-state of water molecules during a 100 ns trajectory for each simulated system. We note that even based on atomistic trajectories determined using MD simulations, there are several possible ways of defining whether an impinging

water molecule is absorbed or reflected and, as a result, different reported $\alpha$-values (ranging from 0.9 to 1.0) even for the flat surface of pure liquid water (Julin et al., 2013; Barclay and Lukes, 2019; Skarbalius et al., 2021; Tsuruta and Nagayama, 2004; Ohashi et al., 2020). To highlight the impacts of curvature, salinity, and organic coatings on $\alpha$ and compare our results with those of other studies, we report normalized values as $\alpha^* = \alpha/\alpha_{0,w}$, where $\alpha_{0,w}$ is the value reported for the flat surface of pure liquid water in the same study. For studies that did not report $\alpha_{0,w}$,

we used $\alpha_{0,w} = 1$ for simplicity. We convert organic concentrations into a surface coating factor, $f_s$, defined as the ratio of the surface area occupied by an idealized monolayer of organic molecules (with the thickness of a single carbon monolayer, 0.35 nm) to the surface area of an idealized spherical water droplet. For consistency, we restrict our comparison to studies that considered semi-volatile or non-volatile organic molecules with chain-like structures and $> 5$ carbon atoms.

Figure 6 shows $\alpha^*$ as a function of $1/D$ based on our results (circles) and previous studies of aerosols containing liquid water and organic matter (stars) (Ma et al., 2011; Chakraborty and Zachariah, 2008; Li et al., 2019; Takahama and Russell, 2011; Diveky et al., 2019; Duffey et al., 2013; Diveky et al., 2021). The $\alpha^*$-values of the droplets





containing pure or salty water ($f_s = 0$) show no apparent dependence on salinity but an increase with droplet size. This size-dependence of $\alpha^*$ is consistent with previous observations that the mass accommodation coefficient of small

water droplets depends on their surface curvature, particularly at droplet diameters below 3 nm (Barclay and Lukes, 2019). Although the origin of this size dependence is unclear, we note that previous studies have demonstrated that the angle of incidence of impinging molecules can significantly impact $\alpha^*$, with more tangential collisions resulting in smaller $\alpha^*$-values (Nagayama and Tsuruta, 2003; Garrett et al., 2006). For an idealized geometry (with point-sized water molecules impacting a perfectly spherical droplet), the distribution of incidence angles of impinging water

vapor molecules should be invariant with droplet size. The size-dependence of $\alpha^*$, suggests that deviations from ideality, such as particle surface roughness or the finite size of incident water molecule, may impact the outcome of particle-water collision events.

For our low organic loading systems ($f_s = 0.30$), shown as yellow circles, calculated $\alpha^*$-values show similar size-dependence as for NaCl droplets and converge to 0.70 in droplets larger than 3 nm. This observation is consistent

with the expectation that for droplets with a sub-monolayer organic coating, the accommodation coefficient can be roughly approximated as the fraction of the droplet surface that is not coated by organic matter, i.e. $\alpha^* \simeq 1 - f_s$, as expected based on the simplistic assumption that $\alpha^* \simeq 1$ on the surface of liquid water and $\alpha^* \simeq 0$ on the organic-coated surface. This relationship between $f_s$ and $\alpha^*$ is also consistent with previous MD simulation results obtained with water droplets containing azelaic acid, a C9-dicarboxylic acid ($f_s = 0.35$, $\alpha^* = 0.60$) (Ma et al., 2011). For our

high organic loading systems ($f_s = 1.50$), shown as orange circles, predicted $\alpha^*$ values are significantly smaller than for our low organic loading systems, as expected, with $\alpha^* \simeq 0.23$ at the largest droplet sizes. Again, our $\alpha^*$ results for high organic loading systems are consistent with values obtained in previous MD simulation and experimental studies for systems with $f_s > 1$, which range from 0 to 0.3 as shown in Figure 6. Interestingly, $\alpha^*$ now decreases with increasing droplet size, displaying a size dependence opposite to that observed for the other systems discussed above.

This unexpected behavior appears to be caused by the distinct size-dependence of the phase-mixing state observed for organic-rich droplets, with smaller droplets displaying an organic-core, water-coated morphology, whereas larger droplets have an 'engulfed' structure that exposes less water at its surface.

## 4 Conclusions

To the best of our knowledge, this work presents an unprecedentedly broad MD simulation examination of the prop-

erties of liquid water in sub-10 nm aerosol particles. Specifically, we characterize droplet shape, water and solute distribution, water diffusivity parallel and normal to the interface, surface tension, water evaporation free energy, water activity, and the water accommodation coefficient in aerosol particles with six different aqueous chemistries for a range of particle sizes. We build upon previous MD simulation studies that have carried out more piecemeal characterizations of water structure, dynamics, and energetics in aerosol particles (Karadima et al., 2019; Chowdhary

and Ladanyi, 2009; Li et al., 2011, 2010). We propose a sphericity factor ($\phi$) that enables the quantitive character-




ization of the mixing structure of water-, organic-, and salt-rich phases and helps reveal the links between aerosol particle morphology and the distinct energetic and kinetic properties of water molecules at nanometer scale.

A key finding of this study is that the size dependence of water microphysics in ultrafine aerosol particles is relatively mild and can be represented using continuum scale models in aerosol particles larger than 4 nm, whereas abrupt changes that require more detailed understanding of molecular level interactions are observed in sub-4 nm diameter particles. For particles consisting of a water-rich phase, smaller droplets display less water cohesion, stronger ion concentration enhancement in the droplet core, faster interfacial water diffusion, smaller surface tension, smaller water accommodation coefficient, and higher water activity. For organic-rich aerosols, a micelle structure (with an organic core and water coating) can form at small particle sizes, which results in distinct water microphysics compared to water-core aerosols at similar sizes.

*Data availability.* All data relevant to this research are available in the maintext and SI Appendix.

## Appendix A: Diffusion coefficient calculation

### A1 Sensitivity to the simulation thermostat

We tested the impact of different thermostat settings – NVT, NVE, NVT with fixed system angular momentum, and Langevin thermostat (LD) with friction constant $\gamma = 0.5$ ps$^{-1}$ (this value results in a friction that is lower than the internal friction of water, while still providing efficient thermostatting) – on the value of the water diffusion coefficient $D_\parallel$ using the Einstein relation for a water droplet with $N_w = 1000$. Resulting plots of water mean square displacement (MSD) versus time are shown in Figure A4. Different symbols in Figure A4 represents the MSD values of water molecules in different 3 Å layers categorized based on distance from the time-averaged Gibbs dividing interface. As shown in Figure A4, the slope of MSD versus time decreases from the interfacial region to the core of the droplet, indicating that $D_\parallel$ decreases from the interfacial region to the core. As discussed in the main text, the time interval $\tau = [2, 6]$ ps was chosen for the calculation of $D_\parallel$, that is $D_\parallel = 1/4 \times \frac{\Delta \text{MSD}}{\Delta \tau} = 1/4 \times \frac{\text{MSD(6 ps)} - \text{MSD(2 ps)}}{4 \text{ ps}}$.

The corresponding $D_\parallel$ values determined for different water layers in simulations carried out with different thermostats are shown in Figure A5. From Figure A5, we can see that the NVT thermostat provides the largest $D_\parallel$ values for all the water layers analyzed, followed by the NVE thermostat. Results obtained using the NVT with fixed angular momentum thermostat are consistent with the results obtained using the LD thermostat and approach the correct bulk-liquid-water diffusion coefficient value $D = 0.25$ Å$^2$ ps$^{-1}$ in the droplet core. The difference in $D_\parallel$ values obtained using different thermostats is consistent with our visual observation of droplet configuration during the MD simulations. In simulations with the NVT thermostat, the simulated isolated droplet shows consistent rotational motion around the droplet's center of mass (com), such that the standard Einstein relation overestimates $D_\parallel$, particularly near the interface. This effect is inhibited in simulations carried out using the NVT with fixed angular





momentum thermostat or the LD thermostat. Given that the bulk-liquid-like region of individual droplets should show $D_\parallel$ values equal to bulk-liquid-water diffusion coefficient, $D = 0.25$ Å$^2$ ps$^{-1}$, diffusion coefficients reported in the main text are based on simulations carried out using the NVT thermostat with fixed system angular momentum.

## A2  Sensitivity to the frame of reference ('com' and 'inst' scheme)

### A2.1  Diffusion coefficient parallel to the interface ($D_\parallel$)

The Einstein relation was used to calculate $D_\parallel$ under both the 'com' and 'inst' schemes, i.e., relative to the average or instantaneous interface. Both schemes are expected to provide mutually consistent results for $D_\parallel$, irrespective of distance from the interface, because capillary waves exist only at the interface and are mostly transverse waves that

should have negligible impact on water displacement parallel to the interface. A comparison of $D_\parallel$ values calculated under both schemes is presented in Figure A6(a) for droplets with $N_w = 3000$. As expected, the two schemes provide mutually consistent values. $D_\parallel$ calculated for different simulated systems under the 'com' scheme are shown in Figure A6(b).

### A2.2  Diffusion coefficient normal to the interface ($D_\perp$)

As discussed in the main text, $D_\perp$ values were determined as a function of distance from the interface by comparing simulation trajectories to the solution of the anisotropic Smoluchowski equation. The 'com' and 'inst' schemes are expected to yield different $D_\perp$ values near the interface (larger in the case of the 'com' scheme), but similar results in the bulk-liquid-like region, because the 'com' scheme intrinsically includes motions due to capillary waves as part of the interfacial water dynamics, whereas the 'inst' scheme excludes the interface fluctuation and only quantifies

water displacement relative to the instantaneous interface. Since the magnitude of capillary waves increases with droplet size (as observed based on both $\phi$ values and interfacial width as noted above), the difference between the $D_\perp$ values at the interface calculated with the two schemes should increase with droplet size.

As shown in Figure A7, we obtained mutually consistent $D_\perp$ values near 0.25 Å$^2$ ps$^{-1}$ under both 'com' and 'inst' schemes for the bulk-liquid-region in the pure liquid water droplet with $N_w = 5000$ ($D = 6.6$ nm), which validates

our calculation method. For the same systems, for the interfacial water layer with a thickness of 3 Å, we obtained a $D_\perp$ value in the range of [0.20, 0.22] Å$^2$ ps$^{-1}$ under the 'com' scheme and a smaller value in the range of [0.15, 0.17] under the 'inst' scheme as shown in Figure A8. We note that although the observation of smaller interfacial $D_\perp$ values under the 'inst' scheme compared with the 'com' scheme is consistent with our expectation, our interfacial water $D_\perp$ values under both schemes are smaller than the values reported in previous simulation studies for the flat

water-air interface (Liu et al., 2004; Wick and Dang, 2005). This discrepancy might be due to different water models, different interfacial layer definitions, or the differences in the curve fitting procedure required to determine $D_\perp$. As noted in the main text, our observations of relatively low $D_\perp$ values near the interface are consistent with reports of



slower water reorientational kinetics near hydrophobic surfaces. The discrepancies between different reported values of $D_\perp$ for interfacial water should be carefully examined in future studies.

**Appendix B: Deliquesence relative humidity as a function of nanoparticle size**

To evaluate the DRH based on our simulation results, we assume that for a nanoparticle at the DRH, ion concentration in the droplet core equals the solubility of the salt $C_{sol}$. We denote the wet particle diameter (including water molecules) as $D_{wet}$ and the cation hydration radius as $r_{hyd}$. If we follow the same assumptions as in Section 3.2 (i.e., that the salt is excluded from the vicinity of the water surface in a region with a thickness equal to 1.2 $r_{hyd}$), the

ion concentration corresponding to a well-mixed system, $C$, can be calculated from

$$1/6\pi D_{wet}^3 C = 1/6\pi(D_{wet} - 2.4 r_{hyd})^3 C_{sol} \tag{B1}$$

The dry diameter of this deliquescent particle $D$ can be approximately evaluated as falling within the range $[D_{salt}, D_{wet}]$, where $D_{salt}$ is the diameter of a pure salt crystal approximated as a sphere with density equal to the bulk solid density. The DRH as a function of $D$, $DRH(D)$ can be calculated from

$$DRH(D) = a_w(C) \tag{B2}$$

And the expression of $a_w(C)$ is taken from the experimental study of bulk-liquid solutions (El Guendouzi and Dinane, 2000).



**Table A1.** Equimolar radius ($Re$), interfacial width ($d_i$), inverse of the sphericity factor ($1/\phi$), and surface tension ($\sigma$) values for water-NaCl mixtures with different ion concentrations ($C$) and for water-pimelic acid droplets with different surface organic loadings ($C_s$) for all simulated systems. $Re$ values were determined from the location of water's Gibbs dividing surface except where denoted with a *, where they were calculated as $[3(N_w/\rho_w + N_{org}/\rho_{org})/4\pi]^{1/3}$ using the bulk water density $34.02 \pm 0.19$ molecules nm$^{-2}$ and bulk organic density $4.373 \pm 0.003$ molecules nm$^{-2}$ calculated from separate MD simulations of each fluid.

| Systems | C | $N_w$ | $d_i$ (Å) | $R_e$ (Å) | $1/\phi$ | $\sigma$ (mN/m) |
|---|---|---|---|---|---|---|
| Water/NaCl | 0 m | 100 | 2.70 | 9.23 | $1.057 \pm 0.007$ | $48.17 \pm 3.31$ |
| | | 500 | 3.29 | 15.13 | $1.064 \pm 0.004$ | $53.04 \pm 3.34$ |
| | | 1000 | 3.56 | 19.11 | $1.067 \pm 0.003$ | $53.91 \pm 2.51$ |
| | | 3000 | 4.36 | 27.66 | $1.070 \pm 0.002$ | $55.08 \pm 2.27$ |
| | | 5000 | 4.70 | 32.83 | $1.071 \pm 0.002$ | $54.92 \pm 2.71$ |
| | 1 m | 500 | 3.12 | 15.24 | $1.063 \pm 0.004$ | $53.32 \pm 3.75$ |
| | | 1000 | 3.27 | 19.27 | $1.065 \pm 0.003$ | $54.66 \pm 3.15$ |
| | | 3000 | 3.65 | 27.83 | $1.067 \pm 0.002$ | $56.08 \pm 2.64$ |
| | | 5000 | 3.81 | 33.03 | $1.068 \pm 0.002$ | $56.26 \pm 3.48$ |
| | 2 m | 500 | 2.80 | 15.40 | $1.063 \pm 0.004$ | $54.01 \pm 3.59$ |
| | | 1000 | 3.14 | 19.43 | $1.064 \pm 0.003$ | $57.08 \pm 3.35$ |
| | | 3000 | 3.50 | 28.06 | $1.067 \pm 0.002$ | $58.09 \pm 3.16$ |
| | | 5000 | 3.83 | 32.29 | $1.068 \pm 0.002$ | $57.83 \pm 4.46$ |
| | 4 m | 500 | ⌒ | 15.42 | $1.296 \pm 0.016$ | $54.55 \pm 5.10$ |
| | | 1000 | ⌒ | 19.50 | $1.303 \pm 0.011$ | $53.32 \pm 3.87$ |
| | | 3000 | ⌒ | 28.28 | $1.347 \pm 0.006$ | $56.92 \pm 3.60$ |
| | | 5000 | ⌒ | 33.58 | $1.337 \pm 0.005$ | $54.86 \pm 4.66$ |
| Water/Pimelic acid | 0.65 PML nm$^{-2}$ | 100 | 5.87 | 8.82 | $1.094 \pm 0.012$ | $17.10 \pm 15.24$ |
| | | 500 | 5.06 | 15.24 | $1.111 \pm 0.009$ | $51.47 \pm 3.77$ |
| | | 1000 | 5.21 | 19.21 | $1.116 \pm 0.007$ | $52.37 \pm 3.69$ |
| | | 3000 | 5.35 | 27.72 | $1.118 \pm 0.005$ | $54.06 \pm 2.49$ |
| | | 5000 | 5.37 | 32.89 | $1.117 \pm 0.004$ | $54.22 \pm 3.55$ |
| | 3.27 PML nm$^{-2}$ | 100 | ⌒ | 13.28* | $1.484 \pm 0.074$ | $34.52 \pm 7.60$ |
| | | 500 | ⌒ | 20.56* | $1.278 \pm 0.031$ | $44.54 \pm 5.09$ |
| | | 1000 | ⌒ | 24.78* | $1.313 \pm 0.021$ | $46.62 \pm 3.69$ |
| | | 3000 | ⌒ | 33.62 * | $1.318 \pm 0.011$ | $49.86 \pm 2.49$ |
| | | 5000 | ⌒ | 38.96* | $1.305 \pm 0.011$ | $50.43 \pm 6.81$ |



**(a)**  **(b)**  **(c)**

**(d)**  **(e)**  **(f)**

**(g)**  **(h)**  **(i)**

**Figure A1.** MD simulation snapshots of droplets with water (gray color), Na$^+$ (dark blue spheres), Cl$^-$ (light blue spheres), and pimelic acid (colored chains with O atoms in red, C atoms in light blue, and H atoms in white): (a) $N_w = 3000$. (b)-(d) 1 m, 2 m, and 4 m NaCl solutions with $N_w = 3000$. (e)-(f) low and high organic loadings with $N_w = 3000$. (g)-(h): low and high organic loadings with $N_w = 100$. Panel (i) shows the same system as panel (h) (high organic loading cluster with $N_w = 100$), but with water represented as a dark blue surface.





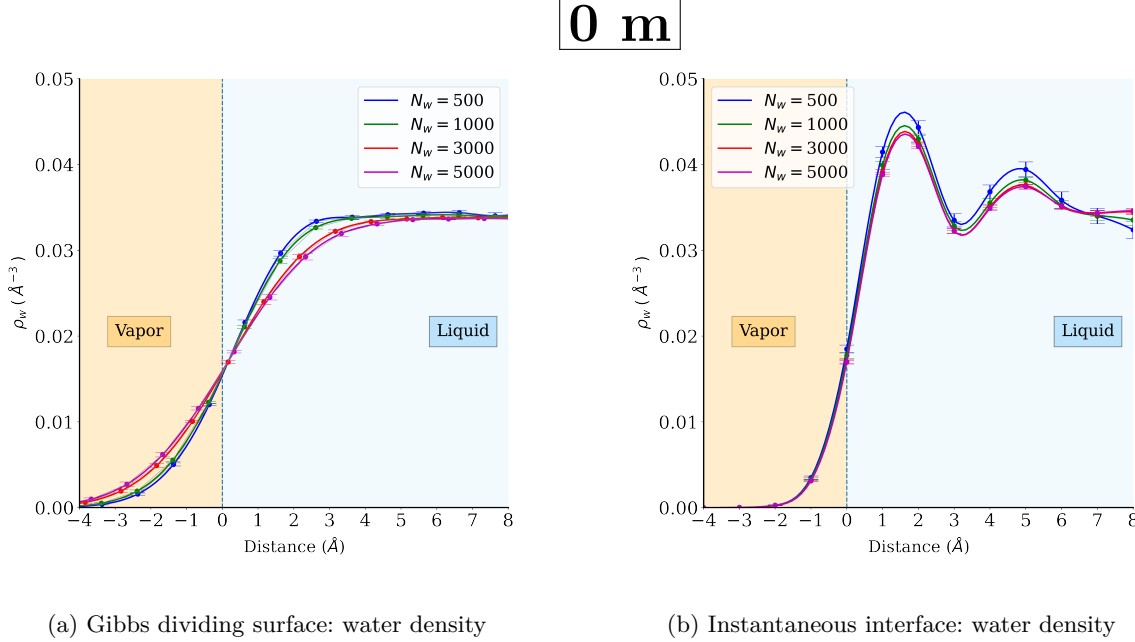

(a) Gibbs dividing surface: water density

(b) Instantaneous interface: water density

**Figure A2.** Atomic density profiles of different components relative to distance from the Gibbs dividing surface (left column) and the instantaneous interface (right column). The top row (a)-(b) shows water density profiles in pure water droplets. Rows 2-3 (c)-(f) show water and ion density profiles in droplets with 1 m NaCl. Rows 4-7 (g)-(n) show results at other aqueous chemistries.



$$\boxed{1 \text{ m}}$$



(c) Gibbs dividing surface: water density

(d) Instantaneous interface: water density

(e) Gibbs dividing surface: ion density

(f) Instantaneous interface: ion density

**Figure A2.** Atomic density profiles of different components relative to distance from the Gibbs dividing surface (left column) and the instantaneous interface (right column). The top row (a)-(b) shows water density profiles in pure water droplets. Rows 2-3 (c)-(f) show water and ion density profiles in droplets with 1 m NaCl. Rows 4-7 (g)-(n) show results at other aqueous chemistries.





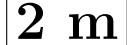

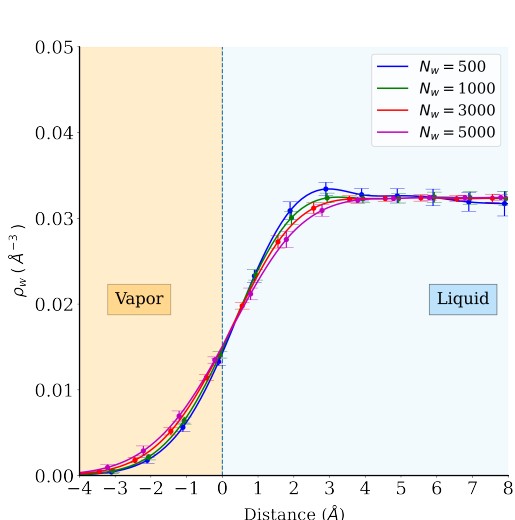

(g) Gibbs dividing surface: water density

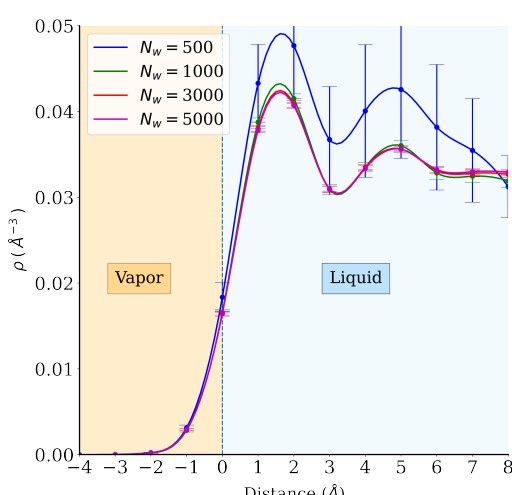

(h) Instantaneous interface: water density

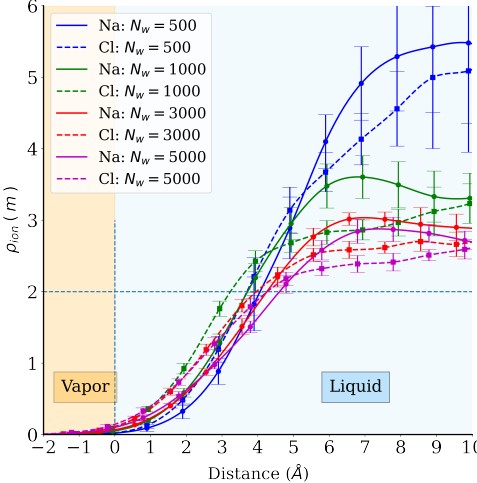

(i) Gibbs dividing surface: ion density

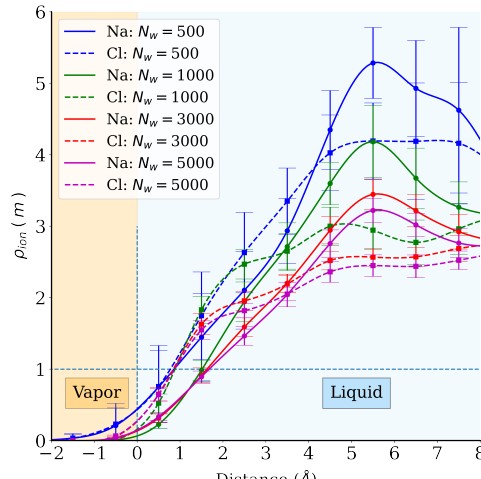

(j) Instantaneous interface: ion density

**Figure A2.** Atomic density profiles of different components relative to distance from the Gibbs dividing surface (left column) and the instantaneous interface (right column). The top row (a)-(b) shows water density profiles in pure water droplets. Rows 2-3 (c)-(f) show water and ion density profiles in droplets with 1 m NaCl. Rows 4-7 (g)-(n) show results at other aqueous chemistries.




low organic loading

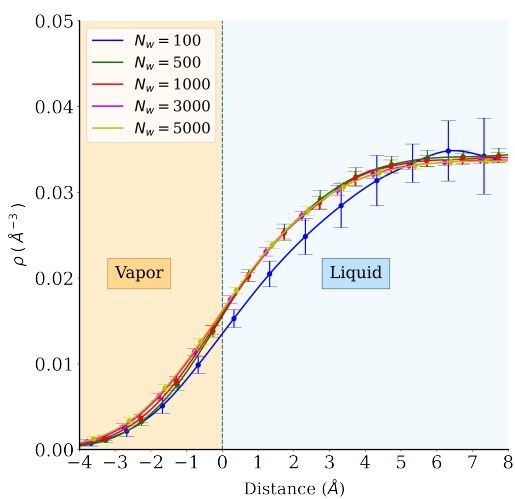

(k) Gibbs dividing surface: water density

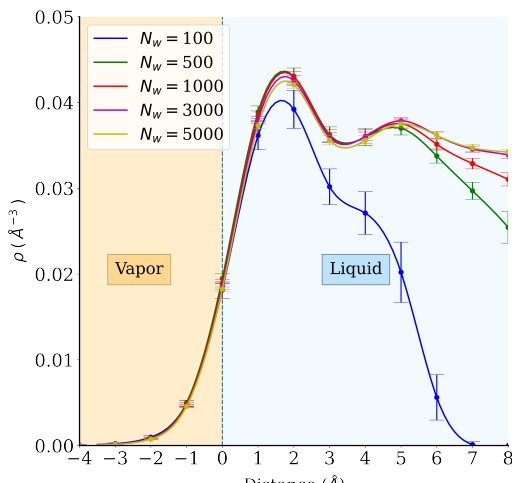

(l) Instantaneous interface: water density

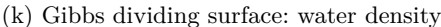

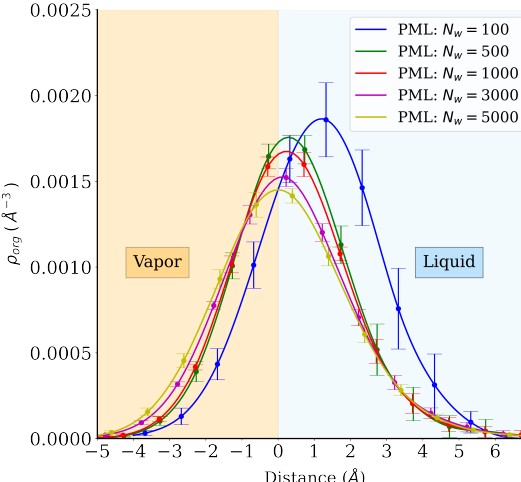

(m) Gibbs dividing surface: pimelic acid (PML) density

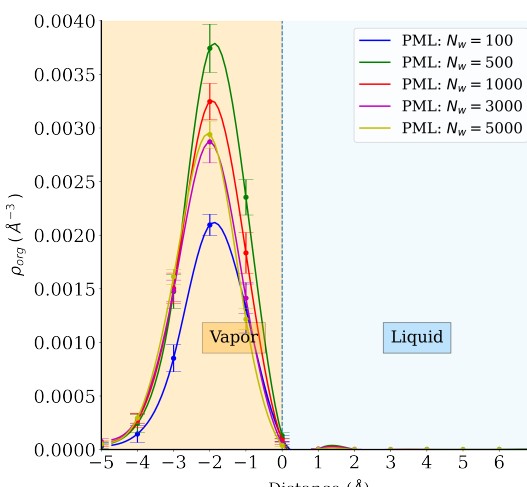

(n) Instantaneous interface: pimelic acid (PML) density

**Figure A2.** Atomic density profiles of different components relative to distance from the Gibbs dividing surface (left column) and the instantaneous interface (right column). The top row (a)-(b) shows water density profiles in pure water droplets. Rows 2-3 (c)-(f) show water and ion density profiles in droplets with 1 m NaCl. Rows 4-7 (g)-(n) show results at other aqueous chemistries.





4 m

(a) Gibbs dividing surface: water density

(b) Gibbs dividing surface: ion density

high organic loading

(c) Gibbs dividing surface: water density

(d) Gibbs dividing surface: pimelic acid (PML) density

**Figure A3.** Atomic density profiles of different components relative to distance from the Gibbs dividing surface for the phase separated systems (4 m NaCl or a high organic loading).





**Figure A4.** Mean square displacement (MSD) of water molecules as a function of time evaluated in different interfacial layers in the direction parallel to the time-averaged Gibbs dividing surface in simulations carried out with different thermostats: NVT (top left), NVE (top right), NVT with fix angular momentum (bottom left), Langevin themostat (LD) with friction constant $\gamma = 0.5$ ps$^{-1}$ (bottom right) for a pure water droplet with $N_w = 1000$.





**Figure A5.** Diffusion coefficient $D_{\parallel}$ of water molecules (i.e., in the direction parallel to the interface) as a function of distance from the interface in simulations of a pure water droplet with $N_w = 1000$ carried out using different thermostats (calculated from the slope, from 2 to 6 ps, of the MSD data shown in Figure A4).




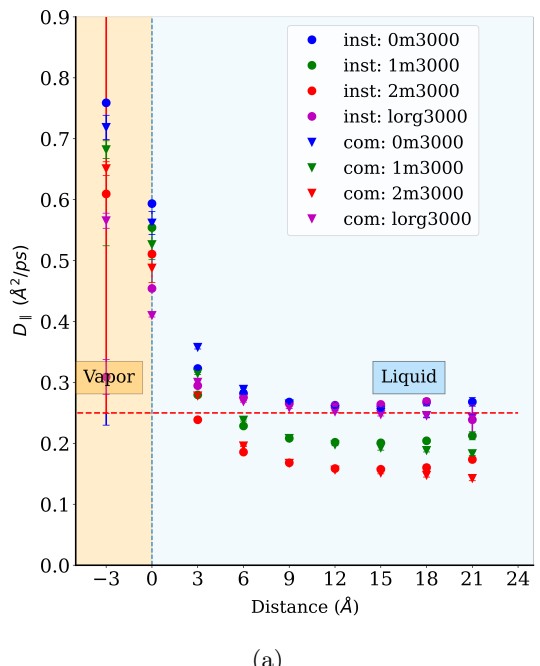
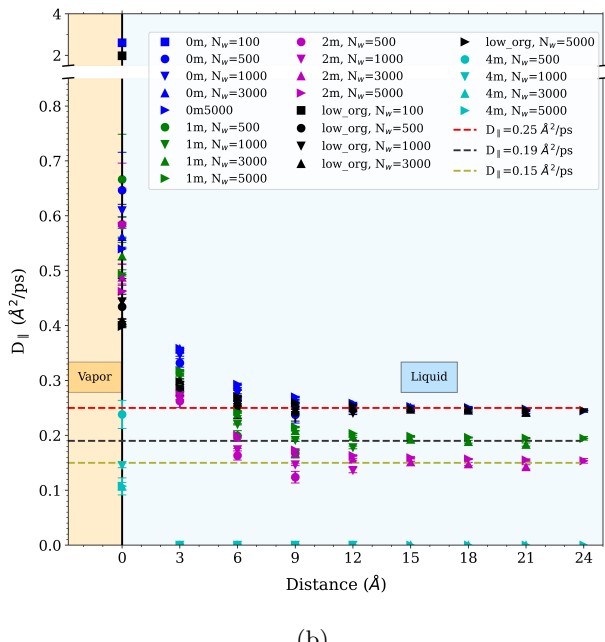

(a)                                    (b)

**Figure A6.** Water diffusion coefficient in the direction parallel to the interface, as a function of distance from the interface: (a) under the center of mass ('com') and the instantaneous interface ('inst') schemes for droplets containing 3000 water molecules with salinity of 0 m, 1 m, and 2 m NaCl, and low organic surface loading ('lorg') and (b) in different simulated systems under the 'com' scheme.





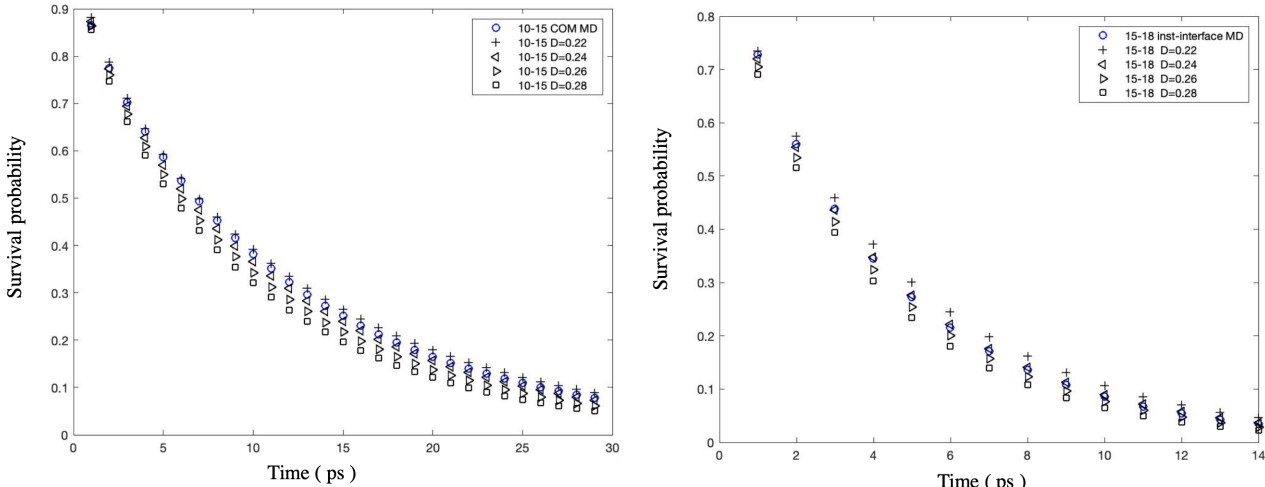

**Figure A7.** Validation of our numerical solution of the anisotropic Smoluchowski equation: (left) survival probability of water molecules as a function of time in a bulk-liquid-like water layer (with distance from the Gibbs dividing surface [1,1.5] nm) in a 5000-water droplet under the 'com' scheme; (right) same calculation for a bulk-liquid like water layer (with distance from the instantaneous interface [1.5,1.8] nm) under the 'inst' scheme. The water diffusion coefficient perpendicular to the interface ($D_\perp$) is calculated by optimizing the agreement between the water survival probability in MD simulations (blue circle) and the Smoluchowski prediction obtained with different $D_\perp$ values (black symbols). The results show that both schemes show consistent results for $D_\perp$ in bulk-liquid droplet regions, with $D_{\perp,\text{bulk}} = (0.22, 0.24)$ Å$^2$ ps$^{-1}$.



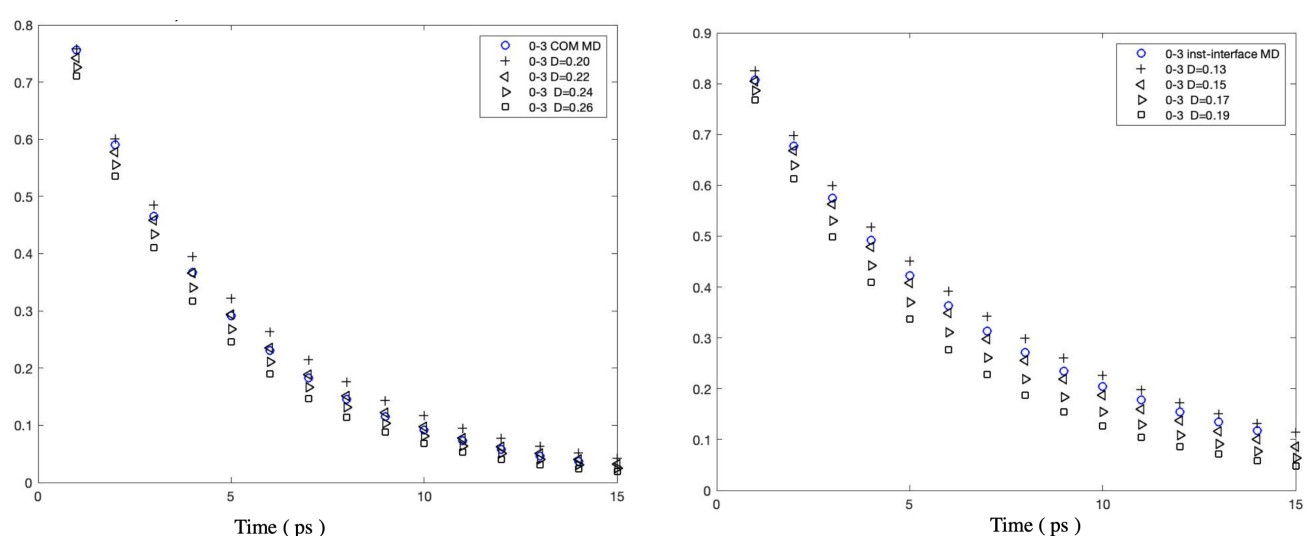

**Figure A8.** Survival probability of water molecules in a 0.3 nm thick interfacial layer in a 5000 water droplet: (left) under the 'com' scheme; (right) under the 'inst' scheme. The results show that $D_{\perp,\text{interface}} = (0.20, 0.22)$ Å$^2$ ps$^{-1}$ under the 'com' scheme, and $D_{\perp,\text{interface}} = (0.15, 0.17)$ Å$^2$ ps$^{-1}$ under the 'inst' scheme.



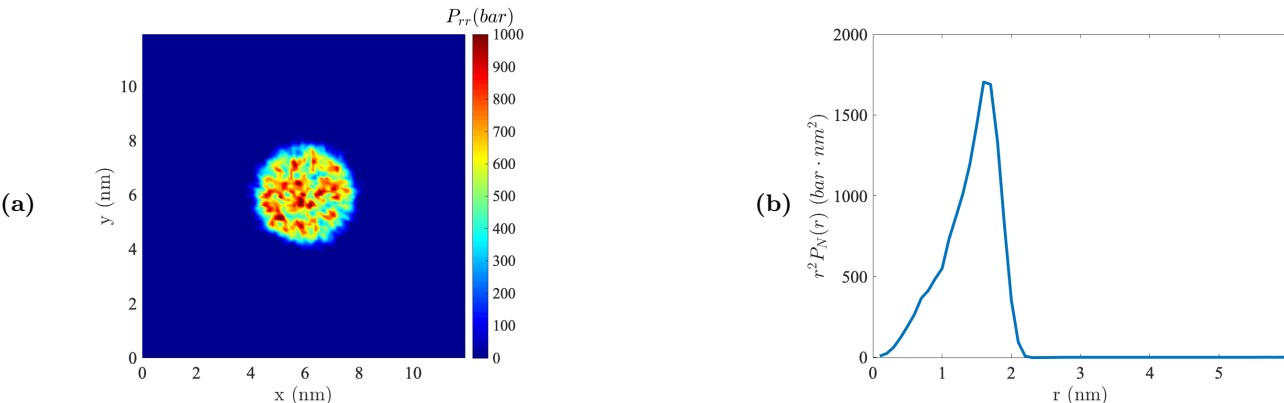

**Figure A9.** (a) Instantaneous normal pressure tensor in a 1000 water droplet at the equatorial plane perpendicular to the z axis. (b) Instantaneous value of $r^2 P_N(r)$ versus radial coordinate in the droplet containing 1000 water molecules.

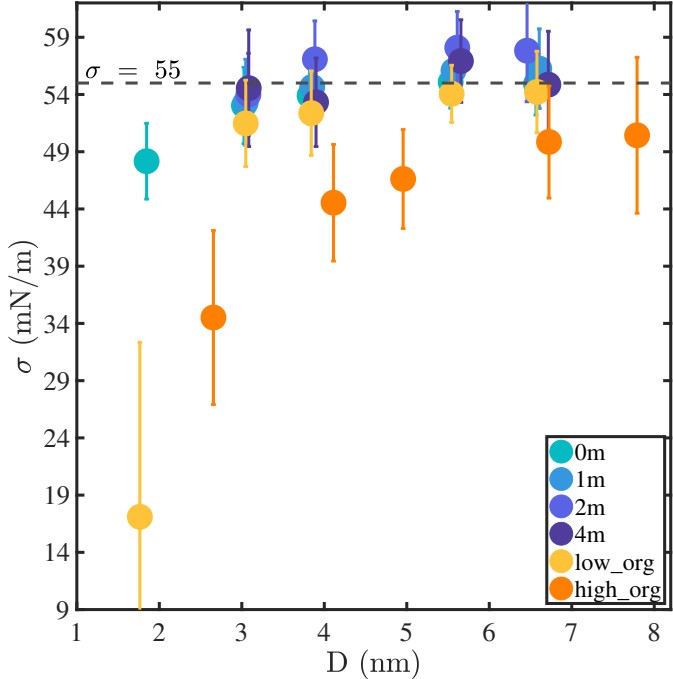

**Figure A10.** Surface tension ($\sigma$) of droplets with different compositions as a function of droplet diameter.





*Author contributions.* X.L. and I.C.B. designed research; X.L. performed research; X.L. analyzed data; X.L. wrote the paper; I.C.B. edited the paper.

*Competing interests.* The authors declare no competing interests.

*Acknowledgements.* This research was supported by the U.S. Department of Energy, Office of Science, Office of Basic Energy Sciences, Geosciences Program under Award DE-SC0018419. Molecular dynamics simulations were performed using computational resources managed and supported by the National Energy Research Scientific Computing Center (NERSC), which is supported by the U.S. Department of Energy, Office of Science, under Award DE-AC02-05CH11231, and by Princeton
Research Computing, a consortium of groups including the Princeton Institute for Computational Science and Engineering (PICSciE) and the Office of Information Technology's High Performance Computing Center and Visualization Laboratory at Princeton University.



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
