# Peer review of "Microphysics of liquid water in sub-10 nm ultrafine aerosol particles"

_EGUsphere, 2022_

## Author Comment (AC1)

**Response to community comments (CC1) by Anthony Wexler**

Dear Professor Wexler,

Thanks for your careful reading of our manuscript entitled "***Microphysics of liquid water in sub-10 nm ultrafine aerosol particles***". We highly appreciate your time and valuable suggestions. Below you will find our replies to your comments.

Best wishes,

Xiaohan Li & Ian C. Bourg
Department of Civil and Environmental Engineering, Princeton University

Summary

This is a comprehensive study of the thermodynamics of nanoscale aerosol particles presenting some unintuitive but well explained results. There are some shortcomings in the work that need to be addressed before it is ready for publication.

Major comments

L13: Relevance to sea spray particles is questionable since (a) particles this small are likely composed primarily of sea surface organics and (b) the physical processes that generated sea spray are not able to generate particles in the ~10nm size range.

Response: Thanks for raising these questions. We agree that sea salt particles are likely to contain organic compounds. However, salt dissolved in seawater, also is a non-negligible component in these particles. Our simulations examine the impact of dissolved salts and organics separately, which cannot reveal salt-organic interactions but facilitates the examination of the impact of each compound. With regard to particle size, previous studies have shown that atmospherically relevant sea salt particles have sizes as small as 10 nm[1,2,3], and recent climate models have employed a mode in the range of 1-10 nm diameter to better characterize the impact of ultrafine sea salt aerosol on cloud formation and climates[4].

L125: what are the implications of a 1.2 nm cut-off for Coulombic and VDW interactions? 1.2 nm is much larger than these molecules and ions and monopole-dipole interactions could be significant. What is the justification for this cut-off.

Response: Thanks for proposing this great question. Here is our response to your questions:

1. **Justification of the cut-off scheme in MD simulations**: The most computationally expensive part of an MD simulation is generally the calculation of nonbonded forces,

including both electrostatic and van der Waals interactions, which act between all pairs of atoms. A general approach to reduce the cost of this computation is to treat electrostatic and van der Waals (VDW) interactions using a well-established cut-off scheme, whereby (1) atom-atom pairwise interactions are explicitly computed within the cut-off distance (i.e., 1.2 nm in our simulation); (2) atom-atom pairwise interactions beyond the cut-off distance is calculated using a reaction-field method combined with shifting or switching functions. Specifically, for electrostatic interactions, we use the particle mesh Ewald sum treatment[5] to divide the interactions into a short-range contribution (distance less than 1.2 nm), and a long-range contribution (distance beyond 1.2 nm). The short-range contribution is calculated in real space, whereas the long-range contribution is calculated using a Fourier transform. While such Ewald methods involve a cutoff distance, the choice of cutoff acts to shift the computational burden between the short-range (real space) and long-range (reciprocal space) calculations, without limiting the accuracy of the calculated forces. For VDW interactions, we accurately characterized the VDW interactions within the 1.2 nm cut-off distance and applied a tail-correction to account for the missing dispersion interactions due to VDW cut-off[6]. In conclusion, all VDW and electrostatic interactions (even beyond the short range cutoff) are accounted for in our simulations.

2. **Justification of the cut-off choice 1.2 nm**: A rule of thumb in determining the cut-off distance in MD simulations is that the cutoff should be larger than $2.5\sigma$ ($\sigma$ is the distance corresponding to the minimum of the VDW pairwise interaction: 0.3166 nm for interactions between two O atoms in $H_2O$, 0.35 nm for C atoms in organic molecules) to accurately characterize the pairwise VDW interactions, but less than half of the simulation cell length to reduce the computation burden and maintain the accuracy of the Ewald summation[7]. There have been extensive studies discussing the choice of cut-off for the interatomic interaction potential models of water, salts, and organic compounds[8,9,10,11,12]. These studies have shown that at ambient temperature, MD simulations using a real space cutoff of 1.2 nm with a particle mesh Ewald sum treatment of long-range Coulomb interactions, can accurately characterize the dynamics, structures, and energetic properties of systems related to those examined in our study compared with experiments.

**Equation 1: How is the interfacial width parameter determined? How sensitive are the results to changes in its value? What value was used?**

Response: Thanks for this great question! The first hydration shell radii ($r_{hyd}$) of alkali cations $Li^+$, $Na^+$, and $K^+$ are 0.279, 0.318, and 0.359 nm, respectively, as reported by Joung and Cheatham[13]. The interfacial width parameter (exclusion width $d_{exc}$) was determined following two steps: (1) from the ion density profile relative to the Gibbs dividing interface (Appendix Figure A2e), we can see that the distance where $Na^+$ density reaches the well-mixed density 1m, ranges from 0.3 to 0.5 nm for all simulated droplets, which provides estimates of the upper and lower bounds of $d_{exc}$; (2) we tested different values of the ion exclusion width and calculated the corresponding ion concentration enhancement factor and, then, compared the calculated results with our MD simulation

Figure 1: Predicted $\epsilon$ of NaCl droplets as a function droplet diameter D with different $d_{exc}$

results. Our results show that setting $d_{exc}$ to $1.2r_{hyd}$, yielded the best match to our MD simulation results. The sensitivity of $\epsilon$ values of NaCl droplets to $d_{exc}$ are illustrated in Figure 1 on the right. We will add this analysis to the Appendix section in the next version of our manuscript.

L450-480: This is the opposite trend to what I expected. Since NaCl is concentrated in the core of the particle due to exclusion from the surface, that should lower the water activity in the particle relative that that in the bulk at the same NaCl concentration. The opposite trend is observed. More discussion about this discrepancy is needed.

Response: Thanks for pointing this out. As noted in our Manuscript L455, water activity can be expressed as

$$a_w = x_w \exp\frac{2\sigma V_m}{RTr}$$

According to this equation, water activity $a_w$ is influenced by two properties: (1) salt concentration, through the term $x_w$ (water mole fraction), and (2) droplet curvature $(1/r)$ through the term $\exp\frac{2\sigma V_m}{RTr}$, which originates from the Kelvin effect. For the salty droplets studied here, the size dependence of $a_w$ is dominated by the Kelvin effect term: as droplet size r decreases, 1/r increases, $\exp\frac{2\sigma V_m}{RTr}$ increases, and $a_w$ increases. The ion concentration enhancement in smaller droplets, acts in the opposite direction, but its impact is comparatively smaller. As noted in our manuscript, under the ideal solution assumption that underlies Köhler theory, 2 m NaCl droplet should have $x_w$ = 0.93, whereas our result shows that nano-droplets considered in this study have an effective water mole fraction of $x_w$ = 0.86 ± 0.02, which indicates that the impact of NaCl on the activity of water is rough twice as large as predicted by Raoult's law. This discrepancy between our results compared with theoretical Köhler theory predictions comes from the ion concentration enhancement in the core of small droplets.

Minor comments

Equation 2: What are N_w and N_org?

Equation 6: what are P_k and P_U?

L174: vapor pressure of the bulk water?

L269: different numbers of water molecules

L299: diminished

L333: dividing

Response: Thanks for your suggestions! We have updated the text accordingly in our newest manuscript.

**References**

(1)  Clarke, A. D.; Owens, S. R.; Zhou, J. An Ultrafine Sea-Salt Flux from Breaking Waves: Implications for Cloud Condensation Nuclei in the Remote Marine Atmosphere. *J. Geophys. Res. Atmospheres* **2006**, *111* (D6). https://doi.org/10.1029/2005JD006565.

(2)  Ahlm, L.; Jones, A.; Stjern, C. W.; Muri, H.; Kravitz, B.; Kristjánsson, J. E. Marine Cloud Brightening – as Effective without Clouds. *Atmospheric Chem. Phys.* **2017**, *17* (21), 13071–13087. https://doi.org/10.5194/acp-17-13071-2017.

(3)  Clarke, A.; Kapustin, V.; Howell, S.; Moore, K.; Lienert, B.; Masonis, S.; Anderson, T.; Covert, D. Sea-Salt Size Distributions from Breaking Waves: Implications for Marine Aerosol Production and Optical Extinction Measurements during SEAS. *J. Atmospheric Ocean. Technol.* **2003**, *20* (10), 1362–1374. https://doi.org/10.1175/1520-0426(2003)020<1362:SSDFBW>2.0.CO;2.

(4)  Pianezze, J.; Barthe, C.; Bielli, S.; Tulet, P.; Jullien, S.; Cambon, G.; Bousquet, O.; Claeys, M.; Cordier, E. A New Coupled Ocean-Waves-Atmosphere Model Designed for Tropical Storm Studies: Example of Tropical Cyclone Bejisa (2013–2014) in the South-West Indian Ocean. *J. Adv. Model. Earth Syst.* **2018**, *10* (3), 801–825. https://doi.org/10.1002/2017MS001177.

(5)  Darden, T.; York, D.; Pedersen, L. Particle Mesh Ewald: An N·log(N) Method for Ewald Sums in Large Systems. *J. Chem. Phys.* **1993**, *98* (12), 10089–10092. https://doi.org/10.1063/1.464397.

(6)  Shirts, M. R.; Mobley, D. L.; Chodera, J. D.; Pande, V. S. Accurate and Efficient Corrections for Missing Dispersion Interactions in Molecular Simulations. *J. Phys. Chem. B* **2007**, *111* (45), 13052–13063. https://doi.org/10.1021/jp0735987.

(7)  Frenkel, D.; Smit, B. *Understanding Molecular Simulation: From Algorithms to Applications*; Elsevier, 2001.

(8)  Mark, P.; Nilsson, L. Structure and Dynamics of the TIP3P, SPC, and SPC/E Water Models at 298 K. *J. Phys. Chem. A* **2001**, *105* (43), 9954–9960. https://doi.org/10.1021/jp003020w.

(9)  Piana, S.; Lindorff-Larsen, K.; Dirks, R. M.; Salmon, J. K.; Dror, R. O.; Shaw, D. E. Evaluating the Effects of Cutoffs and Treatment of Long-Range Electrostatics in Protein Folding Simulations. *PLOS ONE* **2012**, *7* (6), e39918. https://doi.org/10.1371/journal.pone.0039918.

(10) Huang, C.; Li, C.; Choi, P. Y. K.; Nandakumar, K.; Kostiuk, L. W. Effect of Cut-off Distance Used in Molecular Dynamics Simulations on Fluid Properties. *Mol. Simul.* **2010**, *36* (11), 856–864. https://doi.org/10.1080/08927022.2010.489556.

(11) Sonibare, K.; Rathnayaka, L.; Zhang, L. Comparison of CHARMM and OPLS-Aa Forcefield Predictions for Components in One Model Asphalt Mixture. *Constr. Build. Mater.* **2020**, *236*, 117577. https://doi.org/10.1016/j.conbuildmat.2019.117577.

(12) Joung, I. S.; Cheatham, T. E. Determination of Alkali and Halide Monovalent Ion Parameters for Use in Explicitly Solvated Biomolecular Simulations. *J. Phys. Chem. B* **2008**, *112* (30), 9020–9041. https://doi.org/10.1021/jp8001614.

(13) Joung, I. S.; Cheatham, T. E. Molecular Dynamics Simulations of the Dynamic and Energetic Properties of Alkali and Halide Ions Using Water-Model-Specific Ion Parameters. *J. Phys. Chem. B* **2009**, *113* (40), 13279–13290. https://doi.org/10.1021/jp902584c.

---

## Author Comment (AC2)

**Response to community comments (CC2) by Robert McGraw**

Summary: This is a valuable and comprehensive study that models the interactions between water and typical CCN species, that include ions and organics, using molecular dynamics simulation. The most serious reservation that I have concerns Sec. 2.3 and misuse of the terms equimolecular dividing surface (at Re) and surface tension (see below). Unfortunately, Re is used in the equations, beginning with Eq. 3, instead of the "surface of tension", which should be used. This is likely to affect the calculations that follow, especially if the interfacial structure is broad. The authors should have look at this and comment. Otherwise the paper seems important and should be published. Major points, minor points, and a few typos are listed below.

Dear Dr. McGraw,

Thanks for your careful reading of our manuscript entitled "Microphysics of liquid water in sub-10 nm ultrafine aerosol particles". We highly appreciate your time and valuable suggestions. Below you will find our replies to your comments.

Best wishes,

Xiaohan Li & Ian C. Bourg
Department of Civil and Environmental Engineering, Princeton University

**Major**

Section 2.2 System prep and MD simulations: I have some points of confusion after reading this section. The first concerns the underlying model consisting of cubic cells with periodic boundary conditions and edge length exceeding the droplet diameter – a figure here would help the reader.  Second, what is the advantage of periodic boundaries, with so much extra space in each cell?  How is the Ewald sum applied in this model? Usually Ewald sums are applied to extended periodic structures - not to a period set of droplets with space around each one. More details here would be helpful.

Figure 1: 1 m NaCl droplet with 5000 water molecules in a simulation cell of edge length of 21 nm. Simulation cell boundaries are represented as blue lines.

**Response to question 1 (Q1): Visualization of simulation box and droplet.** Thanks for your great suggestion! With regard to the illustration of our simulation cell structure, Figure 1 shows a representative simulation used in our study (specifically, of a droplet with diameter $D \cong 6$ nm in a 21 nm simulation box). We will add this illustration to the supplementary information of our newer version manuscript.

**Response to Q2: Justification of system set-up.** Thanks for raising these questions. The simulation geometries used in our study, including periodic boundary conditions (PBCs), extra space in the simulation cell, and Ewald sum treatment of long-range interactions, are well-established settings and have been widely used in previous MD simulation studies of nano-aerosol droplets[1,2,3,4,5].  In molecular dynamics (MD) simulations, PBCs are usually employed to preserve the thermodynamic and kinetic properties of simulated systems, such as

temperature, pressure, density, and diffusivity[6,7]. Extensive void space is needed for simulations of droplets for two main reasons: (1) to avoid potential artefacts associated with the "corner effect" in cubic simulation cells with PBCs as illustrated in Figure 2a; and (2) to avoid interactions between the droplet and its periodic images as illustrated in Figure 2b. A rule of thumb in MD simulations is that the distance between macromolecules or isolated droplets and their periodic images should equal at least twice the cut-off distance applied to pairwise interatomic interactions (cut-off = 1.2 nm in this study)[7]. In our simulations, simulated droplets were always located at least 3 nm from the simulation cell boundary (and hence > 6 nm from their periodic images). As already examined in previous studies,[1, 2,3,4,5] this should ensure that the thermodynamic and kinetic properties of the simulated droplet are not affected by their periodic images.

With regard to the particle-mesh Ewald summation used in this study, although it was originally introduced as a means to compute the energy of infinite ionic crystals, the Ewald technique is commonly used for MD simulations of non-crystalline or inhomogeneous systems.[8] We have added a sentence to the methods section indicating that the Ewald sum treatment of long-range Coulomb interactions used in our study is commonly used in MD simulations, including in simulations of aerosol particles, and that care should be taken that it introduces artefacts that remain incompletely examined, particularly in the dynamics of charged and dipolar species. For more detail, we refer to the work of Hub et al.[8] and Chapter 12 in Frenkel and Smit[7].

[Figure]

| (a) Illustration of corner effect | (b) Illustration of droplet with its emperical boundary image |

Figure 2: Justification of the simulation set-up

Having called attention to Ewald sums I might point out a clever test that evaluates the accuracy of intermolecular water potentials. This by comparing the computationally relaxed structures with the 3D structure parameters and densities available for ice structures from x-ray diffraction [Morse and Rice, 1981]. For what its worth, the ST2 water potential performed quite well in the test while another did poorly.

**Response to Q3: Choice of interatomic water potentials.** Thanks for proposing this lovely idea. We agree that comparing with experimental results is a nice way to examine the accuracy of interatomic potentials. However, we want to point out that there are different interatomic water models for different condition (e.g. different temperature range). The ST2 water model tends to enhance tetrahedral order and has therefore frequently been used to examine water properties in the supercooled region ($T < 273$ K)[9, 10]. However, it deviates significantly from measured water properties at ambient temperature. For example, ST2 water's density maximum at atmospheric pressure occurs at ~330 K (versus 277 K in actual water).[11] In contrast, the SPC/E water model predicts many properties of liquid water relatively accurately at ambient temperature (e.g. 298.15 K in our study) as already examined in previous studies. With respect to bulk properties at 298K, its radial distribution function is quite accurate, its density is within 1% of experiment, its compressibility of 4.1 × $10^{-10}$ Pa$^{-1}$ is close to the experimental value of 4.5 × $10^{-10}$ Pa$^{-1}$, and its dielectric constant of 70 compares well with the experimental value of 78.2.[12,13, 14] A comparison of 14 water models against synchrotron X-ray data on the atomistic-level structure of liquid water showed that the SPC/E model yields one of the most accurate predictions (about four times more accurate than the ST2 water model).[15] The properties of its vapor-liquid transition are also quite good: the model is explicitly parametrized to reproduce the experimental enthalpy of vaporization, and its vapor pressure is within a factor of 2 of the experimental value.[14] With regards to

transport properties, its self-diffusion coefficient of about 2.5×10$^{-5}$ cm$^2$ /s compares well the experimentally measured value of 2.3×10$^{-5}$ cm$^2$ /s.[16] These properties lead us to believe that the SPC/E model captures sufficient water-like behavior to be useful in our study.

Section 2.3. The author's description of the Gibb's dividing surface seems to this reviewer a misrepresentation of this important concept. Specifically, the authors use of surface tension at the equimolar (equimolecular might be better in context of MD) is said to "correspond to a vanishing adsorption … ensuring that the surface free energy per unit area so defined corresponds to the surface tension". Actually the equimolecular surface does neither! It is the dividing surface located at the "surface of tension" that has these properties. As for adsorption, the Gibbs adsorption isotherm applies only at the surface of tension. Moreover, the pressure difference across the surface of tension is the only one that appears in the standard Laplace and Kelvin relations (otherwise additional terms added to these relations are required) . See [McGraw and Laaksonen, 1997] and especially the citation to Ono and Kondo, an excellent review of the subject, therein.

**Response to Q4: Description of Gibbs dividing surface.** Thank you very much for pointing this out. Actually, the description in our manuscript that "*The density profile was used to locate the equimolar (Gibbs) dividing surface, which corresponds to a vanishing adsorption in a one-component system, ensuring that the surface free energy per unit area so defined corresponds to the surface tension*" was quoted from Lau et al.[17]. We agree, however, that the equimolar surface and the surface of tension are different, and the sentence "*the surface free energy per unit area so defined corresponds to the surface tension*" is not appropriate. Therefore, we will correct our description of equimolar dividing surface (line134-135), as suggested, to "*The density profile was used to locate the equimolar dividing surface, which corresponds to a vanishing adsorption in a one-component system.*".

Related: Eq. 7 is similar to the equation developed by Gibbs for the work to form a capillary drop from vapor. This formula can be applied even to droplets having a broadened interfacial region - provided the radius at the surface of tension is used.

**Response to Q5: Discussion of using equimolar radius $Re$ into surface tension calculation.** Thanks for pointing this out. We agree that the surface tension should be rigorously calculated at the radius Rs, where the surface tension applies, instead of Re. We have modified the phrasing in our manuscript to highlight this distinction and to point out that our use of Re (instead of Rs) is an approximation justified by the following reasoning: (1) For theoretical analysis, the impact of the difference between Re and Rs on the $\sigma$ calculation is small in most of our simulated systems. The Tolman length $\delta$ can be approximated as a measurement of the difference between (Re-Rs). Previous studies, including MD simulations, theoretical analysis, and experimental measurement, have reported $\delta$ values range from -0.1 to 0.2 nm for liquid water systems (e.g. water droplets, bubbles, water cavitation in minerals and water with hydrophobic solute).[18,19,20,21,22,23,24,25] Among these studies, Kim and Jhe[20] directly calculated (Re-Rs) from MD simulations of pure water droplet with radii ranging from 0.5 to 1.5 nm. Their results show that for nanodroplets with radii larger than 0.9 nm, (Re-Rs) is in the range of [-0.05, 0.05] nm. Our results in Section 3.4 similarly show that $\delta$ is very small (~0.06 nm). The sensitivity of Eq. 7 to the choice of R value should be given by the expression:

$$f_{err} = \frac{\frac{d\sigma}{dR}}{\sigma}(R_s - R_e) = -\frac{2R^{-3}}{R^{-2}}(R_s - R_e) = -\frac{2}{R}(R_s - R_e)$$

For $(R_s - R_e) < 0.06$ nm, which is the case in our study, the maximum error introduced by the use of $R_e$ in $\sigma$ calculation is less than 8% for droplets with $N_w > 500$. We have added a sentence to our manuscript noting this potential source of systematic error and, also, noting that it is commensurate with the statistical error of our predicted $\sigma$ values (as shown by our reported error bars of ~$\pm$8%). (2) In MD simulation practice, as noted in our manuscript, the droplets simulated here do not have a perfectly spherical shape. As a result, it is impossible to accurately determine the surface tension surface Rs in a traditional manner. The surface tension calculation scheme used in our study (Eq. 6- Eq. 8), which uses Re for approximation, is widely used in previous MD

simulation studies for nano-droplet surface tension calculation[26,27,3,28,29]. Finally, as will be shown in our next paper, the surface tension values calculated in our study are consistent with experimental measurements on the dependence of droplet surface tension dependence on organic concentration.

Finally, a couple of comments on the "validity of the Kelvin and Kohler theory at droplet sizes larger than 4nm under moderate salinities and organic loadings and the need to account for ion-concentration enhancement in sub-10nm particles" mentioned in the Abstract. This is an important theme that runs through and adds value to the paper. With respect to the Kelvin relation this has been confirmed for the Kelvin (pure water) and Kelvin-Thomson (ionic solution) relations [Winkler et al., 2012]. For Kohler theory, on the other hand, this is unlikely to be the case for organics. The latter tend to partition between the bulk and surface phases, whereas the standard Kohler and kappa-Kohler models pertain only to fully water-soluble species. A recent extension of Kohler theory, based on analysis of droplet stability, takes into account the partitioning of both water-soluble and surface-active species in a unified way for applications to cloud activation [McGraw and Wang, 2021].

**Response to Q6: Discussion about the concept of organic solubility at nanoscale.** Thanks for the nice comments. We agree that the standard Köhler and $\kappa$- Köhler theory were developed for water-soluble organics. However, we would point out that "water-soluble species" is a concept that is well-defined for bulk solutions but not for sub-10 nm droplet, because of the surface partitioning of organic molecules. For example, oxalic acid ($C_2H_2O_4$), is often defined as water-soluble species (solubility in water 118g/L at 25 °C). Experimental studies have shown that the surface excess $\Gamma$ (mol/m$^2$) of oxalic acid can be well-described by Langmuir adsorption isotherm[30] by

$$\Gamma = \frac{\Gamma_\infty C}{b + C}$$

where $\Gamma_\infty$ is the maximum surface excess, $1.54 \times 10^{-6}$ mol/m$^2$; b is the inverse Langmuir adsorption constant, 1.67 mol/L; and C is the bulk solute concentration (mol/L). For a droplet with diameter $D = 10$ nm and with $C = 1$ mol/L, the organic mass ratio of the interfacial and in the droplet can be estimated as

$$\frac{m_{surface}}{m_{bulk}} = \frac{4\pi\Gamma(D/2)^2}{4\pi/3(D/2)^3C} = 0.35$$

As can be seen from the above derivation, in sub-10 nm nanodroplet, the surface partitioning of "water-soluble organic species" is non-negligible anymore. In our study, we compare our simulation results for PML-water clusters with Köhler theory to examine how water activity deviates from Köhler theory predictions under "water-soluble-organic-species" assumption, because this has not previously been examined (to the best of our knowledge). As noted in our manuscript, at low organic loading and droplet larger than 4 nm, the water activity results calculated from our study match the Köhler theory prediction. This is why we are saying that Köhler theory seems valid at moderate organic surface loadings.

McGraw, R. and A. Laaksonen (1997), J. Chem. Phys. 106, 5284-5287.

Morse, M. D. and S. A. Rice (1981), J. Chem. Phys. 74, 6514-6516.

Winkler, P. M., et. al. (2012), Phys. Rev. Letts. 108, 085701.

McGraw, R. and J. Wang (2021), J. Chem. Phys. 154, 024707; doi: 10.1063/5.0031436

**Response to Q7: Comments on the references.** We really appreciate your recommendations of these references. In particular, we think the papers by McGraw, R. and A. Laaksonen (1997) and McGraw, R. and J.

Wang (2021) are highly relevant and helpful to our studies. We will cite these papers in our discussion and conclusion.

Minor points and typos:

Eq. 9 (previously just below Eq. 5) rho_0 was used, which I assume is the density at the center of the drop. Why the switch to rho_w, which I assume is the bulk density of water? I don't see these symbols defined.

The switch from molecular units, kT, to moler units, RT, in equation 9 and back to kT in Eq. 10 can be avoided using consistent units.

Line 777. The correct authorciting should be to Lewis and Schwartz, 2004. Same in line 102: change Lewis et. al. to Lewis and Schwartz, 2004.

**Response to Q8: Comments of minor points.** Thank you so much for these corrections. We will update our newest manuscript accordingly.

**Reference**

(1) Chowdhary, J.; Ladanyi, B. M. Molecular Dynamics Simulation of Aerosol-OT Reverse Micelles. *J. Phys. Chem. B* **2009**, *113* (45), 15029–15039.
(2) Karadima, K. S.; Mavrantzas, V. G.; Pandis, S. N. Insights into the Morphology of Multicomponent Organic and Inorganic Aerosols from Molecular Dynamics Simulations. *Atmospheric Chem. Phys.* **2019**, *19* (8), 5571–5587.
(3) Sun, L.; Hede, T.; Tu, Y.; Leck, C.; Ågren, H. Combined Effect of Glycine and Sea Salt on Aerosol Cloud Droplet Activation Predicted by Molecular Dynamics Simulations. *J. Phys. Chem. A* **2013**, *117* (41), 10746–10752.
(4) Julin, J.; Shiraiwa, M.; Miles, R. E.; Reid, J. P.; Pöschl, U.; Riipinen, I. Mass Accommodation of Water: Bridging the Gap between Molecular Dynamics Simulations and Kinetic Condensation Models. *J. Phys. Chem. A* **2013**, *117* (2), 410–420.
(5) Bahadur, R.; Russell, L. M. Effect of Surface Tension from MD Simulations on Size-Dependent Deliquescence of NaCl Nanoparticles. *Aerosol Sci. Technol.* **2008**, *42* (5), 369–376.
(6) Abraham, M. J.; Murtola, T.; Schulz, R.; Páll, S.; Smith, J. C.; Hess, B.; Lindahl, E. GROMACS: High Performance Molecular Simulations through Multi-Level Parallelism from Laptops to Supercomputers. *SoftwareX* **2015**, *1*, 19–25.
(7) Frenkel, D.; Smit, B. *Understanding Molecular Simulation: From Algorithms to Applications*; Elsevier, 2001.
(8) *Quantifying Artifacts in Ewald Simulations of Inhomogeneous Systems with a Net Charge | Journal of Chemical Theory and Computation*. https://pubs.acs.org/doi/full/10.1021/ct400626b (accessed 2022-11-14).
(9) *Low-temperature fluid-phase behavior of ST2 water: The Journal of Chemical Physics: Vol 131, No 10*. https://aip.scitation.org/doi/10.1063/1.3229892 (accessed 2022-11-14).
(10) *Two-state thermodynamics of the ST2 model for supercooled water: The Journal of Chemical Physics: Vol 140, No 10*. https://aip.scitation.org/doi/10.1063/1.4867287 (accessed 2022-11-14).
(11) Harrington, S.; Poole, P. H.; Sciortino, F.; Stanley, H. E. Equation of State of Supercooled Water Simulated Using the Extended Simple Point Charge Intermolecular Potential. *J. Chem. Phys.* **1997**, *107* (18), 7443–7450. https://doi.org/10.1063/1.474982.
(12) Vega, C.; McBride, C.; Sanz, E.; Abascal, J. L. F. Radial Distribution Functions and Densities for the SPC/E, TIP4P and TIP5P Models for Liquid Water and Ices Ih, Ic, II, III, IV, V, VI, VII, VIII, IX, XI and XII. *Phys. Chem. Chem. Phys.* **2005**, *7* (7), 1450–1456. https://doi.org/10.1039/B418934E.
(13) Mark, P.; Nilsson, L. Structure and Dynamics of the TIP3P, SPC, and SPC/E Water Models at 298 K. *J. Phys. Chem. A* **2001**, *105* (43), 9954–9960. https://doi.org/10.1021/jp003020w.
(14) Varilly, P.; Chandler, D. Water Evaporation: A Transition Path Sampling Study. *J. Phys. Chem. B* **2013**, *117* (5), 1419–1428.

(15) *Water structure as a function of temperature from X-ray scattering experiments and ab initio molecular dynamics - Physical Chemistry Chemical Physics (RSC Publishing)*. https://pubs.rsc.org/en/content/articlelanding/2003/cp/b301481a (accessed 2022-11-17).

(16) Tsimpanogiannis, I. N.; Moultos, O. A.; Franco, L. F. M.; Spera, M. B. de M.; Erdős, M.; Economou, I. G. Self-Diffusion Coefficient of Bulk and Confined Water: A Critical Review of Classical Molecular Simulation Studies. *Mol. Simul.* **2019**, *45* (4–5), 425–453. https://doi.org/10.1080/08927022.2018.1511903.

(17) Lau, G. V.; Ford, I. J.; Hunt, P. A.; Müller, E. A.; Jackson, G. Surface Thermodynamics of Planar, Cylindrical, and Spherical Vapour-Liquid Interfaces of Water. *J. Chem. Phys.* **2015**, *142* (11), 114701.

(18) Min, S. H.; Berkowitz, M. L. Bubbles in Water under Stretch-Induced Cavitation. *J. Chem. Phys.* **2019**, *150* (5), 054501. https://doi.org/10.1063/1.5079735.

(19) Wilhelmsen, Ø.; Bedeaux, D.; Reguera, D. Communication: Tolman Length and Rigidity Constants of Water and Their Role in Nucleation. *J. Chem. Phys.* **2015**, *142* (17), 171103.

(20) Kim, Qh.; Jhe, W. Interfacial Thermodynamics of Spherical Nanodroplets: Molecular Understanding of Surface Tension via a Hydrogen Bond Network. *Nanoscale* **2020**, *12* (36), 18701–18709. https://doi.org/10.1039/D0NR04533K.

(21) Burian, S.; Isaiev, M.; Termentzidis, K.; Sysoev, V.; Bulavin, L. Size Dependence of the Surface Tension of a Free Surface of an Isotropic Fluid. *Phys. Rev. E* **2017**, *95* (6), 062801. https://doi.org/10.1103/PhysRevE.95.062801.

(22) Menzl, G.; Gonzalez, M. A.; Geiger, P.; Caupin, F.; Abascal, J. L. F.; Valeriani, C.; Dellago, C. Molecular Mechanism for Cavitation in Water under Tension. *Proc. Natl. Acad. Sci.* **2016**, *113* (48), 13582–13587. https://doi.org/10.1073/pnas.1608421113.

(23) Azouzi, M. E. M.; Ramboz, C.; Lenain, J.-F.; Caupin, F. A Coherent Picture of Water at Extreme Negative Pressure. *Nat. Phys.* **2013**, *9* (1), 38–41. https://doi.org/10.1038/nphys2475.

(24) Joswiak, M. N.; Duff, N.; Doherty, M. F.; Peters, B. Size-Dependent Surface Free Energy and Tolman-Corrected Droplet Nucleation of TIP4P/2005 Water. *J. Phys. Chem. Lett.* **2013**, *4* (24), 4267–4272. https://doi.org/10.1021/jz402226p.

(25) *The spontaneous curvature of the water-hydrophobe interface: The Journal of Chemical Physics: Vol 137, No 13*. https://aip.scitation.org/doi/full/10.1063/1.4755753 (accessed 2022-08-26).

(26) Li, X.; Hede, T.; Tu, Y.; Leck, C.; Ågren, H. Surface-Active Cis-Pinonic Acid in Atmospheric Droplets: A Molecular Dynamics Study. *J. Phys. Chem. Lett.* **2010**, *1* (4), 769–773.

(27) Zhao, Z.; Kong, K.; Wang, S.; Zhou, Y.; Cheng, D.; Wang, W.; Zeng, X. C.; Li, H. Understanding Hygroscopic Nucleation of Sulfate Aerosols: Combination of Molecular Dynamics Simulation with Classical Nucleation Theory. *J. Phys. Chem. Lett.* **2019**, *10* (5), 1126–1132.

(28) Li, X.; Hede, T.; Tu, Y.; Leck, C.; Ågren, H. Cloud Droplet Activation Mechanisms of Amino Acid Aerosol Particles: Insight from Molecular Dynamics Simulations. *Tellus B Chem. Phys. Meteorol.* **2013**, *65* (1), 20476.

(29) Zakharov, V. V.; Brodskaya, E. N.; Laaksonen, A. Surface Tension of Water Droplets: A Molecular Dynamics Study of Model and Size Dependencies. *J. Chem. Phys.* **1997**, *107* (24), 10675–10683.

(30) Aumann, E.; Hildemann, L.; Tabazadeh, A. Measuring and Modeling the Composition and Temperature-Dependence of Surface Tension for Organic Solutions. *Atmos. Environ.* **2010**, *44* (3), 329–337.

---

## Author Comment (AC3)

**Response to reviewer comments (RC2)**

The authors present a theoretical study of nanoparticle morphology and gas/droplet partitioning behavior of water using systems consisting of sodium chloride, water, and pimelic acid. The authors discover several parameters - sphericity and fractional surface coverage - that aptly describe chemical morphology as a function of composition and size regimes and variation in mass accommodation coefficients. The authors also report a threshold for the validity of continuum theories. The paper is well-written and is of interest to the Atmospheric Chemistry and Physics community, and is recommended for publication after the following general comments have been addressed.

Dear Reviewer,

Thanks for your careful reading of our manuscript entitled "**Microphysics of liquid water in sub-10 nm ultrafine aerosol particles**". We highly appreciate your time and valuable suggestions. Below you will find our replies to your comments.

Best wishes,

Xiaohan Li & Ian C. Bourg
Department of Civil and Environmental Engineering, Princeton University

As the authors note in Section 3.4, classical water models are known to have biases in errors in reproducing experimental surface tensions - though with SPC/E having one of the smallest errors (Vega and de Miguel, 2007). Additionally, a study (Lbadaoui-Darvas and Takahama, 2019) suggest that carboxylic acid-water dynamics are not well captured in equilibrium MD simulations and lead to deviations in predictions of water activity even above 0.95. On the other hand, the water activity calculations seem to suggest that the simulation results are in good agreement - with observations - is this due to canceling of errors (e.g., with molar volume) or the relatively small magnitude of the error in surface tension by these models?

**Response to question 1 (Q1): Discussion about the water activity results.** Thanks for raising this question. First of all, we would like to point out that the calculations of water activity ($a_w$) in dicarboxylic acid-water systems carried out by Lbadaoui-Darvas and Takahama[1] differ from those reported in our manuscript in several important ways: (1) we use different interatomic models to simulate carboxylic-acid-water systems; more precisely, we use the OPLS-AA model for dicarboxylic acid and the SPC/E model for water, whereas they used the OPLS-UA model for dicarboxylic acid and the TIP4P model for water; (2) our systems include a water-air interface, where dicarboxylic acid molecules preferentially accumulate relative to bulk liquid water, whereas they simulate organics in bulk liquid water; and (3) we calculated water activity using two different methods (umbrella sampling and the vapor-liquid coexistence method), whereas they calculated water activity using a third method (from the derivative of chemical potential vs. solute mole fraction, where the chemical potential was inferred from structural results). Therefore, differences between our water activity predictions are not unexpected.

More importantly, the calculations reported by Lbadaoui-Darvas and Takahama exclusively evaluate the Raoult effect in bulk aqueous solutions (i.e., the decrease in water activity associated with the organic solute). They find that this effect is underestimated by their MD simulations, with a resulting error in predicted water activity of up to about 5%. Our simulations, however, quantify the overall impact of both the Raoult and Kelvin effects in

nano-scale droplets. Unfortunately, we cannot precisely quantify the Raoult effect, for two reasons: (1) the Kelvin effect is much larger than the Raoult effect in our simulated systems (the expected Kelvin enhancement is up to ~250%; the expected Raoult inhibition is only up to ~10%), and (2) the bulk aqueous concentration of dicarboxylic acid in our systems is not well defined, because we study nanoscale droplets with significant organic accumulation at the interface, not bulk aqueous solutions.

We agree that SPC/E water model used in our study should underestimate the Kelvin effect, because it somewhat underestimates water surface tension (although less than most other water models , as noted by the reviewer). In our comparison with the Kelvin equation and Köhler theory, we account for this discrepancy by using the surface tension of SPC/E water, instead of that of real water, to obtain Kelvin/Köhler theory predictions of water activity.

Many of the conclusions summarize the effect of "organic loadings" but the simulations use a specific type of organic, namely pimelic acid. Many studies on the other hand suggest the importance of alcohols in marine aerosols (e.g., Russell et al., 2010). Is there reason that the authors can justify broadening the conclusion from a particular "organic acid" to "organics" generally? Other abundant dicarboxylic acids (e.g., oxalic acid) may also exhibit different bulk/surface partitioning behavior than demarcated by the sphericity factor. The main question is whether parts of the manuscript should be more clear in what is meant by "organic loading" in this work.

**Response to question 2 (Q2): The choice of organic species and clarification of organic loading.** Thank you for raising this important point. We justify our usage of pimelic acid (PML) as a representative of general aerosol organic compounds due to the following reasons: (1) organic species with 6 or 7 carbon and 3 or 4 oxygen atoms are important in organic aerosols: for example, online and offline spectrometer measurements of $PM_{2.5}$ in Beijing by Zheng et al. (2021)[2] revealed that the carbon number of organic species in organic aerosols was distributed from 2 to 20, with the highest abundance observed for C6-C7 species, and that organics most commonly contained 3 or 4 oxygen atoms. This finding is also validated by measurements above temperate and boreal forests, where the molecular weight of organic species was predominantly in the range of 150-200 g mol$^{-1}$ (for reference, the molecular weight of PML is 160 g mol$^{-1}$). [3] Furthermore, C3-C11 dicarboxylic acid (DA) species have been shown to be important contributors to total organic aerosol mass and can contribute ~50% of the total DA mass both in urban and rural areas.[4,5] (2) As noted in our manuscript, the O/C ratio of PML (0.57) lies near the midpoint of the range commonly observed for aerosol organic materials (0.2 to 1.0) and near the value below which liquid-liquid phase separation is commonly observed in aerosol particles ($\sim$ 0.7 to 0.8 for organic-salt-water aerosols, $\sim$ 0.6 for organic-water aerosols). Therefore, we believe PML has the potential to mimic the properties of key organic substances in nano-aerosol droplets. However, we agree that a variety of other organic species, including alcohols, other dicarboxylic acids, etc, are also present in aerosols, and that their behavior and impact on aerosol properties may differ from those evidenced for pimelic acid in our simulations (although some generality of our results is suggested, for example, by the consistency of our predicted water accommodation coefficients with values observed in other studies with different organic species, see Figure 6 in our manuscript). To clarify this, we plan to add the following paragraph to our conclusion section:

"Finally, we reiterate that our simulations use of a highly simplified proxy for aerosol organic matter as a single compound (PML). This compound was selected for its similarity to the compounds most abundantly observed in organic aerosols in terms of molecular weight (Thornton et al., 2020), number of C atoms (Zheng et al., 2021), O:C ratio (Song et al., 2018, Zheng et al., 2021), and functional groups (Zhao et al., 2018, Wang et al., 2022). Although the use of such an idealized proxy facilitates our effort to evaluate the sensitivity of aerosol properties to organic loading, future studies should examine whether our predictions can be generalized to other organic compounds (or mixtures of organic compounds) abundantly found in organic aerosols, such as alcohols or other dicarboxylic acids."

References:

Lbadaoui-Darvas, Mária, and Satoshi Takahama. "Water Activity from Equilibrium Molecular Dynamics Simulations and Kirkwood-Buff Theory." The Journal of Physical Chemistry B 123, no. 50 (December 19, 2019): 10757–68. https://doi.org/10.1021/acs.jpcb.9b06735.

Russell, L. M., L. N. Hawkins, A. A. Frossard, P. K. Quinn, and T. S. Bates. "Carbohydrate-like Composition of Submicron Atmospheric Particles and Their Production from Ocean Bubble Bursting." Proceedings of the National Academy of Sciences of the United States of America 107, no. 15 (2010): 6652–57. https://doi.org/10.1073/pnas.0908905107.

Vega, C., and E. de Miguel. "Surface Tension of the Most Popular Models of Water by Using the Test-Area Simulation Method." The Journal of Chemical Physics 126, no. 15 (2007): 154707. https://doi.org/10.1063/1.2715577.

**References:**

(1)  Lbadaoui-Darvas, M.; Takahama, S. Water Activity from Equilibrium Molecular Dynamics Simulations and Kirkwood-Buff Theory. *J. Phys. Chem. B* **2019**, *123* (50), 10757–10768. https://doi.org/10.1021/acs.jpcb.9b06735.

(2)  Zheng, Y.; Chen, Q.; Cheng, X.; Mohr, C.; Cai, J.; Huang, W.; Shrivastava, M.; Ye, P.; Fu, P.; Shi, X.; Ge, Y.; Liao, K.; Miao, R.; Qiu, X.; Koenig, T. K.; Chen, S. Precursors and Pathways Leading to Enhanced Secondary Organic Aerosol Formation during Severe Haze Episodes. *Environ. Sci. Technol.* **2021**, *55* (23), 15680–15693. https://doi.org/10.1021/acs.est.1c04255.

(3)  Thornton, J. A.; Mohr, C.; Schobesberger, S.; D'Ambro, E. L.; Lee, B. H.; Lopez-Hilfiker, F. D. Evaluating Organic Aerosol Sources and Evolution with a Combined Molecular Composition and Volatility Framework Using the Filter Inlet for Gases and Aerosols (FIGAERO). *Acc. Chem. Res.* **2020**, *53* (8), 1415–1426. https://doi.org/10.1021/acs.accounts.0c00259.

(4)  Zhao, W.; Kawamura, K.; Yue, S.; Wei, L.; Ren, H.; Yan, Y.; Kang, M.; Li, L.; Ren, L.; Lai, S.; Li, J.; Sun, Y.; Wang, Z.; Fu, P. Molecular Distribution and Compound-Specific Stable Carbon Isotopic Composition of Dicarboxylic Acids, Oxocarboxylic Acids and $\alpha$-Dicarbonyls in $PM_{2.5}$ from Beijing, China. *Atmospheric Chem. Phys.* **2018**, *18* (4), 2749–2767. https://doi.org/10.5194/acp-18-2749-2018.

(5)  Qi, W.; Wang, G.; Dai, W.; Liu, S.; Zhang, T.; Wu, C.; Li, J.; Shen, M.; Guo, X.; Meng, J.; Li, J. Molecular Characteristics and Stable Carbon Isotope Compositions of Dicarboxylic Acids and Related Compounds in Wintertime Aerosols of Northwest China. *Sci. Rep.* **2022**, *12* (1), 11266. https://doi.org/10.1038/s41598-022-15222-6.

---

## Author Response (AR2)

**Response to reviewer comments**

Dear Reviewer,

Thanks for your careful reading of our manuscript entitled "**Microphysics of liquid water in sub-10 nm ultrafine aerosol particles**". We highly appreciate your time and valuable suggestions. Below you will find our replies to your comments.

Best wishes,

Xiaohan Li & Ian C. Bourg
Department of Civil and Environmental Engineering, Princeton University

Thank you for the clarifications of your revised manuscript and the detailed responses. Please address the following additional comments from Referees:
1. Regarding the findings of Zheng et al. 2021 (and similar works), it should be noted that signal intensities from mass spectra are not necessarily the same as molecular abundance since the ionization efficiencies across molecules can vary greatly.

2. Additionally, the results clearly depend on the bulk/surface partitioning behavior of the organics as this appears to determine the agreement in water activity between simulation and experiment - and this partitioning behavior also varies greatly across organic molecules found in atmospheric particles. Therefore, the conclusions might be overstated in present form and described more accurately if written in terms of PML rather than "organic loading" more generally.

**Response to comment 1 and 2: Mass spectra signal intensity and statement of PML or "organic loading".**
Thanks for the great comments. We agree with you that mass spectrometry results are not strictly quantitative due to differences in ionization efficiency of different species. We also agree with you that the phrase of "PML" is more appropriate than "organic loading" to avoid overstatement of our conclusions. We have resolved these concerns in our newest manuscript by (i) rephrasing "organic loading" as "PML loading"; (ii) noting that mass spectrometry results are not strictly quantitative; (iii) noting that additional studies are required to evaluate the sensitivity of our predictions to the type of salt or organic solutes. All the changes regarding comment 1 and 2 are marked as yellow shaded areas in the track-change document.

3. The justification for PML as a model compound is based on studies of aerosol with low contributions of salt, and therefore the relevance of mixtures of PML and NaCl to marine aerosol is not clear. Please provide specific citations to why PML is representative of organic aerosol that has been found to be associated with sea salt.

**Response to question 3 (Q3): PML and sea salt aerosols.** Thanks for the comments. As noted on page 4 of our manuscript, we selected NaCl and PML as representative inorganic and organic solutes frequently observed in different types of aerosol particles. We do not claim that PML is abundant in sea spray aerosol particles, and we simulate NaCl- and PML-containing aerosols separately in our study. As part of our edits, we noted the importance of examining aerosols containing other salts or organic solutes (or mixtures of solutes) in future studies.